# Multi-model Online Conformal Prediction with Graph-Structured Feedback

**Erfan Hajihashemi**                                                      *ehajihas@uci.edu*
*Department of Electrical Engineering and Computer Science*
*University of California, Irvine*

**Yanning Shen**\*                                                        *yannings@uci.edu*
*Department of Electrical Engineering and Computer Science*
*University of California, Irvine*

**Reviewed on OpenReview:** *https://openreview.net/forum?id=9u8ugbismg*

## Abstract

Online conformal prediction has demonstrated its capability to construct a prediction set for each incoming data point that covers the true label with a predetermined probability. To cope with potential distribution shift, multi-model online conformal prediction has been introduced to select and leverage different models from a preselected candidate set. Along with the improved flexibility, the choice of the preselected set also brings challenges. A candidate set that includes a large number of models may increase the computational complexity. In addition, the inclusion of irrelevant models with poor performance may negatively impact the performance and lead to unnecessarily large prediction sets. To address these challenges, we propose a novel multi-model online conformal prediction algorithm that identifies a subset of effective models at each time step by collecting feedback from a bipartite graph, which is refined upon receiving new data. A model is then selected from this subset to construct the prediction set, resulting in reduced computational complexity and smaller prediction sets. Additionally, we demonstrate that using prediction set size as feedback, alongside model loss, can significantly improve efficiency by constructing smaller prediction sets while still satisfying the required coverage guarantee. The proposed algorithms are proven to ensure valid coverage and achieve sublinear regret. Experiments on real and synthetic datasets validate that the proposed methods construct smaller prediction sets and outperform existing multi-model online conformal prediction approaches.

## 1 Introduction

Machine learning models are rapidly improving in providing accurate predictions; however, it remains challenging to ensure that decisions inferred from these models are reliable. To address this challenge, uncertainty quantification can be used to ensure model reliability by providing interval predictions instead of point estimates (Heskes, 1996; Patel, 1989). Reliable interval prediction is particularly important in safety-critical applications such as autonomous driving (Doula et al., 2024; Dixit et al., 2023) and healthcare (Lu et al., 2022; Boger et al., 2025; Vazquez & Facelli, 2022).

Conformal prediction is a model-agnostic and distribution-free uncertainty quantification framework that constructs prediction sets of candidate output values such that the true output is covered in the set with a predefined probability (Vovk et al., 2005). Conventional conformal prediction algorithms achieve the desired coverage under the assumption of data exchangeability—i.e., a sequence of random variables whose joint distribution remains invariant under any permutation of the indices (Balasubramanian et al., 2014). The exchangeability assumption often fails to hold in practice, particularly in online settings where data is

---

\*Corresponding author

collected sequentially. Such violations can lead to a failure to maintain the desired coverage. To address this, several lines of work have focused on online conformal prediction (Zaffran et al., 2022; Feldman et al., 2022). In the online conformal setting, a parameterized prediction set is constructed at each timestep. After observing the true label, the parameters are updated based on the algorithm's performance, such as whether the prediction set included the true label. Adaptive conformal prediction has also been introduced, where prediction sets are constructed in a time-varying manner to cope with the challenges of online environments (Gibbs & Candès, 2021). Even though adaptive conformal prediction algorithms can cover the true label with the desired probability, they may construct inefficient prediction sets (e.g., excessively large sets). The efficiency of conformal prediction algorithms depends on the underlying learning model, and a single model may not perform consistently well across all sequential data.

To address this issue, (Hajihashemi & Shen, 2024; Gasparin & Ramdas, 2024a) proposed leveraging multiple learning models to provide diverse candidates for adaptive conformal prediction algorithms to select the appropriate model. However, their approaches are limited to a set of candidate models with good performance. The learning models used in their experiments are all high-performing, which may not simulate real-world scenarios where lower-performing models are also present in the candidate set. Additionally, multi-model conformal prediction methods can suffer from high computational complexity when the number of candidate models is large, as it requires updating the adaptive conformal prediction parameters for each individual model. To address these limitations, we introduce a new multi-model online conformal prediction algorithm, which simulates the online model selection as graph-structured feedback. The proposed method dynamically selects a subset of effective learning models and prunes weak ones at each time step by constructing a graph. In addition, we propose an extension to the aforementioned algorithm, where the prediction set size of the effective models is used as feedback. This extension enables the construction of significantly smaller prediction sets while still achieving valid coverage of the true label, outperforming previously proposed multi-model conformal prediction approaches.

**Related work:** Conformal prediction is a powerful framework for uncertainty quantification that has been widely used to predict a set of candidate outcomes for input data (Shafer & Vovk, 2008; Vovk, 2015; Papadopoulos et al., 2002). The goal is to quantify the uncertainty of a given black-box machine learning model by constructing a prediction set. Conformal prediction frameworks can be utilized on both classification (Shi et al., 2013; Romano et al., 2020; Ding et al., 2023) and regression (Romano et al., 2019; Boström et al., 2017; Papadopoulos et al., 2011) tasks. Conformal prediction algorithms can be broadly categorized into split and full variants (Barber et al., 2023). In split conformal prediction, the training data is divided into two disjoint subsets: a proper training set and a calibration set. The proper training set is used to fit the point prediction model, while the calibration set is used to compute nonconformity scores (Oliveira et al., 2024). In contrast, full conformal prediction is significantly more computationally demanding, as it requires retraining or scoring the point prediction model for each test point and every possible candidate label (Angelopoulos et al., 2020). Hence, in this work, we only focus on split conformal prediction algorithms.

Employing standard conformal prediction in online environments, where the exchangeability assumption may be violated, does not achieve the desired coverage guarantee. To address this, (Gibbs & Candès, 2021) introduced the use of a time-varying miscoverage probability. However, this approach has certain limitations (e.g., the need to specify the learning rate in advance (Podkopaev et al., 2024)). (Zaffran et al., 2022; Gibbs & Candès, 2024) use expert learning techniques (Vovk, 1995; Cesa-Bianchi et al., 1997; Littlestone & Warmuth, 1994) to mitigate these limitations. (Lei & Candès, 2021; Tibshirani et al., 2019; Podkopaev & Ramdas, 2021) utilized reweighting techniques to cope with changes in online settings. However, these methods often rely on some distributional assumptions. Despite achieving valid coverage guarantees, these methods may fail to construct efficient prediction sets that are both small and able to cover the true label. Some recent works propose using multiple learning models to enhance conformal prediction (Gasparin & Ramdas, 2024a;b; Yang & Kuchibhotla, 2025; Bhagwat et al., 2025; Hajihashemi & Shen, 2024). Both (Yang & Kuchibhotla, 2025; Bhagwat et al., 2025) leverage multiple models in the full conformal prediction setting, which suffers from high computational cost. (Gasparin & Ramdas, 2024a) proposes a majority-vote strategy for aggregating conformal sets in the split conformal prediction setting. However, their method suffers from coverage loss and lacks a theoretical guarantee of achieving the desired $1-\alpha$ coverage. (Hajihashemi & Shen,

2024) proposed selecting a model from a set of candidate models at each time step. This approach can incur high computational cost and reduced efficiency when the set includes poorly performing candidates.

**Contributions.** Overall, our contributions can be summarized as follows:

**I)** We introduce a novel multi-model online conformal prediction algorithm, **G**raph-structured feedback **M**ultimodel Ensemble **O**nline **C**onformal **P**rediction (GMOCP), designed for online environments. At each time step, the algorithm selects a learning model to construct the prediction set from a subset of effective models identified using a graph structure.

**II)** An adaptive framework is proposed for generating the graph based on the performance of each learning model over previous time steps. It is proven that GMOCP achieves sublinear regret and guarantees valid coverage.

**III)** Experiments on real and synthetic datasets demonstrate the effectiveness of the GMOCP method in constructing more efficient prediction sets with lower computational complexity, while achieving a coverage probability closely aligned with the target value.

## 2 Preliminaries

This section provides preliminaries on standard conformal prediction and adaptive conformal prediction. Given a learning model $m$ and a set of historical data $\{(X_\tau, Y_\tau^{\text{true}})\}_{\tau=1}^{t-1}$, where $X_\tau \in \mathcal{X}$ denotes the input data at time $\tau$ and $Y_\tau^{true} \in \mathcal{Y}$ is its corresponding true label, conformal prediction aims to construct a prediction set $C_\alpha^m(X_t) \subseteq \mathcal{Y} := \{1, 2, \ldots, N_{labels}\}$ for a new data $X_t$. Here, $N_{labels}$ denotes the total number of unique labels. The prediction set $C_\alpha^m(X_t)$ is constructed such that it includes the true label $Y_t^{\text{true}}$ with probability $1 - \alpha$, where $\alpha$ denotes the given miscoverage probability. Upon receiving the true label $Y_t^{\text{true}}$, the new data pair $(X_t, Y_t^{\text{true}})$ is added to the historical dataset. In the online setting, conformal prediction uses the evolving historical data as the calibration set to decide which candidate labels should be included in the prediction set. The decision to include each candidate label is based on a threshold determined based on the calibration set. For each datum $X_\tau$ and its corresponding true label $Y_\tau^{\text{true}}$, a non-conformity score $S^m(X_\tau, Y_\tau^{\text{true}})$ is calculated based on the learning model $m$. This score represents the disagreement between the ground-truth label $Y_\tau^{\text{true}}$ and predicted label $\hat{f}^m(X_\tau)$. A lower non-conformity score indicates a better match between the true label and the predicted label by model $m$. Upon calculating non-conformity scores for the entire historical dataset, $\{S^m(X_\tau, Y_\tau^{\text{true}})\}_{\tau=1}^{t-1}$, threshold $\hat{q}_\alpha^m$ is obtained by:

$$\hat{q}_\alpha^m = Quantile\left(\frac{\lceil t(1-\alpha)\rceil}{t-1}, \{S^m(X_\tau, Y_\tau^{true})\}_{\tau=1}^{t-1}\right), \tag{1}$$

where $Quantile(\cdot, \cdot)$ sorts the nonconformity scores in ascending order and then outputs $\frac{\lceil t(1-\alpha)\rceil}{t-1}$ empirical quantile of sorted scores. Next, prediction set for new data $X_t$ is constructed as

$$C_\alpha^m(X_t) = \{Y \in \mathcal{Y} \mid S^m(X_t, Y) \le \hat{q}_\alpha^m\} \tag{2}$$

In online settings, employing a time-invariant miscoverage probability $\alpha$ to determine the threshold $\hat{q}_\alpha^m$ may not achieve the desired coverage guarantee. To address this issue, adaptive conformal prediction has been developed, where a time-variant miscoverage probability $\alpha_t$ is utilized instead of a fixed $\alpha$ to obtain the desired coverage. By replacing $\alpha$ with it's time-variant version $\alpha_t$ in equation 1, the time-varying threshold $\hat{q}_{\alpha_t}^m$ can be updated accordingly.

Even though employing time-variant miscoverage probability is useful in online environments, relying on a single learning model across all time steps may be suboptimal. To address this issue, the use of multiple learning models is considered in previous work (Hajihashemi & Shen, 2024). At each time step, based on the performance of every learning model $m \in [M]$ over previous time steps, the model $\hat{m}$ is selected, and the prediction set is constructed according to threshold $\hat{q}_\alpha^{\hat{m}}$. However, among $M$ candidate learning models, some may exhibit poor performance due to, e.g., insufficient training data. Including such models may result in inefficiently large prediction sets. Moreover, employing a large number of learning models increases the computational complexity. To address these limitations, the present work develops a data-driven approach to select a subset of effective models at each time step, and then choose $\hat{m}$ from this subset.

## 3    Methodology

A data-driven algorithm, **G**raph-structured feedback **M**ultimodal Ensemble **O**nline **C**onformal **P**rediction (GMOCP), which adaptively selects subsets of learning models, is detailed in Subsection 3.1. Subsection 3.2 then discusses the construction of the graph used to identify effective models at each time step. Finally, Subsection 3.3 introduces **E**fficient GMOCP (EGMOCP), which incorporates prediction set size as feedback to further reduce the size of the constructed prediction sets while maintaining coverage guarantees.

### 3.1    Data-driven Model Selection

To address the high computational complexity and inefficiency of large prediction sets, our first approach is proposed, in which a subset of models is adaptively selected 'on the fly' upon receiving new data samples. To adaptively select a subset of effective learning models, our proposed approach utilizes feedback from a graph that is generated in an online fashion based on the performance of each learning model in previous time steps. By doing this, the proposed approach avoids including learning models with weak performance in the candidate set for the conformal prediction task. The details of feedback graph construction will be presented in subsection 3.2.

Consider a time-variant bipartite graph $G_t$ (Asratian et al., 1998), which includes two sets of nodes: $M$ model nodes $\{v_1^{(l)}, ..., v_M^{(l)}\}$ and $J$ selective nodes $\{v_1^{(s)}, ..., v_J^{(s)}\}$ where $v_m^{(l)}$ and $v_j^{(s)}$ represents $m-$th learning model and $j$-th selective node respectively. The edges of the graph represent associations between model nodes and selective nodes. Increasing the number of model nodes connected to $v_j^{(s)}$ can lead to higher computational complexity. Therefore, the graph generation approach should impose a limitation on the maximum number of model nodes connected to each selective node. In this work, each selective node is connected to at most $N$ model nodes. Given $G_t$ at each time step $t$, one selective node is chosen and its associated model nodes, forming a subset denoted by $S_t$, are used as the candidate set for the conformal prediction task. These selected model nodes can contribute to the conformal prediction task either through a weighted sum or by selecting a single model according to a probability mass function (PMF). In this work, we focus on the latter approach. The selection is guided by the weight $w_t^m$ assigned to each model $m \in [M]$, which influences both the generation of the graph $G_t$ and the selection of model $\hat{m}$ from the subset of effective models $S_t$. Specifically, we normalize the weights of models in $S_t$, as $\bar{w}_t^m = \frac{w_t^m}{\sum_{\bar{m} \in S_t} w_t^{\bar{m}}}, \forall m \in S_t$. Then, a model is selected to create the prediction set according to the PMF defined by the normalized weights $\boldsymbol{w}_t^s = (w_t^{\bar{m}})_{\bar{m} \in S_t}$.

After creating the prediction set at time $t$, the true label $Y_t^{true}$ is observed. The threshold $\hat{q}_\alpha^m$ can be obtained according to equation 1 based on non-conformity score functions, where each score function depends on a specific learning model $m$. Given that different learning models yield different non-conformity scores, using a single adaptive miscoverage probability $\alpha_t$ for all learning models is inadequate. To address this, at each time step $t$ we assign a specific miscoverage probability $\alpha_t^m$ to each learning model $m \in [M]$. Since the cardinality of $S_t$ is at most $N$, there are at most $N$ distinct miscoverage probabilities, each updated independently. To update miscoverage probability $\alpha_t^m$ for each $m \in [S_t]$, we adopt the pinball loss defined as (Koenker & Bassett, 1978):

$$L(\bar{\alpha}_t^m, \alpha_t^m) = \alpha(\bar{\alpha}_t^m - \alpha_t^m) - \min\{0, \bar{\alpha}_t^m - \alpha_t^m\}, \tag{3}$$

where

$$\bar{\alpha}_t^m := \sup\{\tilde{\alpha} : Y_t^{true} \in C_{\tilde{\alpha}}^m(X_t)\} \tag{4}$$

is the best possible value of miscoverage probability for model $m$ at time $t$, which constructs the smallest prediction set that covers $Y_t^{true}$. The miscoverage probability $\alpha_{t+1}^m$ can be updated via scale free online gradient descent (SF-OGD) (Orabona & Pál, 2018) as

$$\alpha_{t+1}^m = \alpha_t^m - \eta \frac{\nabla_{\alpha_t^m} L(\bar{\alpha}_t^m, \alpha_t^m)}{\sqrt{\sum_{\tau=1}^t \|\nabla_{\alpha_\tau^m} L(\bar{\alpha}_\tau^m, \alpha_\tau^m)\|_2^2}}, \tag{5}$$

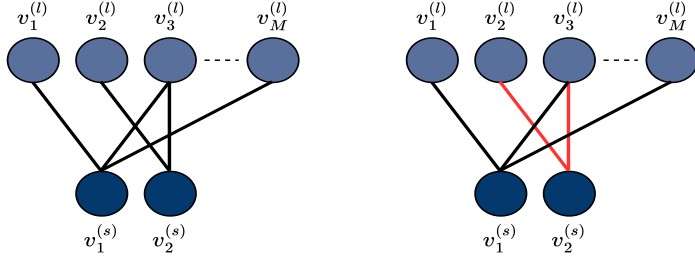

Figure 1: (left): An illustrative example of the generated bipartite graph $G_t$ with $M$ learning models and $J = 2$ selective nodes. (right) The selective node $v_2^{(s)}$ is chosen, and the subset $S_t$ includes all learning models connected to the selected node, highlighted by red edges.

which follows an online gradient descent update with a time-dependent decaying learning rate. The parameter $\eta$ is the learning rate and

$$\nabla_{\alpha_t^m} L(\bar{\alpha}_t^m, \alpha_t^m) = \mathbb{I}[\bar{\alpha}_t^m < \alpha_t^m] - \alpha = err_t^m - \alpha, \tag{6}$$

with $err_t^m := \mathbb{I}[Y_t^{true} \notin C_{\alpha_t^m}^m] = 1$ if the predicted set does not contain the true label $Y_t^{true}$, and 0 otherwise. According to equation 5, the adaptive miscoverage probability is increased when the prediction set includes the true label. This allows the prediction set to become smaller by excluding unnecessary labels $\mathcal{Y}' := \{Y' \in \mathcal{Y} \mid \hat{q}_{\bar{\alpha}_t^m}^m < S^m(X_t, Y') \leq \hat{q}_{\alpha_t^m}^m\}$ in next step. Conversely, if the prediction set fails to include the true label, the adaptive miscoverage probability is decreased.

Additionally, the weights $w_t^m$ for $m \in S_t$ are updated after observing the true label $Y_t^{true}$ by leveraging a multiplicative update rule:

$$w_{t+1}^m = w_t^m \exp\left(-\epsilon l_t^m / 2^b\right), \tag{7}$$

where $\epsilon$ is the step size that controls weight update, $b = \lfloor \log_2 J \rfloor$ and $l_t^m$ denotes the importance sampling loss estimates (Alon et al., 2017)

$$l_t^m = \frac{L\left(\bar{\alpha}_t^m, \alpha_t^m\right)}{q_t^m} \mathbb{I}\{m \in S_t\}, \tag{8}$$

where $q_t^m$ is the probability that the learning model $m$ is included in $S_t$, which depends on how the graph $G_t$ is generated.

Then, a weight $u_{t+1}^j$ is assigned to each selective node $j \in [J]$ according to the model nodes' weights $w_{t+1}^m$. Specifically, $u_{t+1}^j$ is calculated as the sum of the weights $w_{t+1}^m$ of all model nodes connected to the selective node $j$, as follows:

$$u_{t+1}^j = \sum_{\forall m: v_m^{(l)} \to \in v_j^{(s)}} w_{t+1}^m. \tag{9}$$

Moreover, the probability according to which a selective node is chosen in the next time step, denoted by $p'^j_{t+1}$, can be updated as $p'^j_{t+1} = \frac{u_{t+1}^j}{\sum_{i=1}^J u_{t+1}^i}$. To sum up, at each time step, all model nodes connected to the selected selective node form a subset of candidate learning models. This approach aims to avoid including low-performing models in the candidate set for the conformal prediction task. One model is then selected from this subset to construct the prediction set.

## 3.2 Online Graph Generation

The generation of $G_t$ impacts both the selection of candidate learning models for the conformal prediction and the computational complexity. A well-designed graph should lead to the selection of a selective node that is connected to a subset of model nodes, which construct small prediction sets while still covering the true label. Let $A_t$ represent the $M \times J$ sub-adjacency matrix between two disjoint subsets $\{v_1^{(l)}, ..., v_M^{(l)}\}$ and

$\{v_1^{(s)}, ..., v_J^{(s)}\}$. The entry $A_t(m, j)$ denotes the $m-$th row and $j-$th column of matrix $A_t$, and it's value is 1 if there is edge between model node $m$ and selective node $j$ in bipartite graph $G_t$; otherwise it is 0. The probability of connecting model node $v_m^{(l)}$ to each selective node is denoted by $p_t^m$ and can be obtained as:

$$p_t^m = (1 - \eta_e) \frac{w_t^m}{\sum_{\bar{m}=1}^M w_t^{\bar{m}}} + \frac{\eta_e}{M}. \tag{10}$$

The second term in equation equation 10 allows exploration across all model nodes. Specially, each model node is connected to a selective node $v_j^{(s)}$ uniformly at random if $\eta_e = 1$. Each selective node $v_j^{(s)}$ draws model nodes in $N$ independent trials. In each trial, the selective node draws one model node according to PMF $\boldsymbol{p}_t = (p_t^m)_{m=1}^M$. According to the definition of $p_t^m$ in equation 10, the probability that the $m-$th model node is connected to the $j-$th selective node is $1 - (1 - p_t^m)^N$, where $(1 - p_t^m)^N$ represents the probability that $m$th model node is not selected by $j-$th selective node in any of $N$ trials. Hence, the probability that the learning model $m$ is included in $S_t$ is given by

$$q_t^m := \sum_{j=1}^J p'_t^j \left(1 - (1 - p_t^m)^N\right), \tag{11}$$

for all $m \in M$, and is used for importance sampling loss estimate in equation 8. The entire process for generating the graph $G_t$ is detailed in Algorithm 1. Given the graph $G_t$ at each time step, one selective node is chosen according to the PMF $\boldsymbol{p}'_t = (p'_t^j)_{j=1}^J$, where $p'_t^j = \frac{u_t^j}{\sum_{i=1}^J u_t^i}$. Figure 1 illustrates an example of the constructed bipartite feedback graph and the selective node selection process. By considering the model nodes connected to the selected selective node as the set of candidates, one learning model is then selected according to the PMF $\boldsymbol{w}_t^s = (w_t^{\bar{m}})_{\bar{m} \in S_t}$ to construct the prediction set. The entire GMOCP method is summarized in Algorithm 2. The per-iteration cost at time $t$ for GMOCP is $\mathcal{O}(Nt + JMN)$, where the $JMN$ term accounts for graph generation and the $Nt$ term arises from computing $\bar{\alpha}_t^m$ for up to $N$ selected models, using sorted calibration scores of length $t$. The per-iteration complexity of MOCP is $\mathcal{O}(Mt)$. Thus, GMOCP can reduce the per iteration complexity, especially in settings where $N << M$.

---

**Algorithm 1** Generating Graph $G_t$

---

**Require:** Number of selective nodes $J$, exploration coefficient $\eta_e > 0$, the maximum number of connected models to each selective node $N$, $M$ pre-trained models.

  Initialize $A_t = 0_{J \times M}$.
  **for** $m = 1, ..., M$ **do**
    Set $p_t^m = (1 - \eta_e) \frac{w_t^m}{\sum_{\bar{m}=1}^M w_t^{\bar{m}}} + \frac{\eta_e}{M}$.
  **end for**
  **for** $j = 1, ..., J$ **do**
    **for** $n = 1, ..., N$ **do**
      Select one of the models according to PMF $\boldsymbol{p}_t = (p_t^m)_{m=1}^M$
      Set $A_t(j, \tilde{m}) = 1$ {$\tilde{m}$ is selected model from PMF}
    **end for**
  **end for**

---

Let $CovE(T) := \left| \frac{1}{T} \sum_{t=1}^T \mathbb{E}[err_t] - \alpha \right|$ represent the coverage error. The expected error is calculated as $\mathbb{E}[err_t] = \sum_{j=1}^J \bar{u}_t^j \sum_{q=1}^{Q=\binom{M+N-1}{M}} N! \left( \prod_{m=1}^M \frac{(p_t^m)^{b_{m,q}}}{b_{m,q}!} \right) \sum_{m \in S_{t,q}} \bar{w}_t^{mq} err_t^m$ where the expectation is over every possible subset of effective models $S_t$, (see Appendix A.2 for full definition of $b_{m,q}$). Note that here $\bar{w}_t^{mq} = \frac{w_t^m}{\sum_{\bar{m} \in S_{t,q}} w_t^{\bar{m}}}$. The following theorem demonstrates that GMOCP has bounded coverage error (See proof in A.1).

**Theorem 1** *The coverage error of the GMOCP algorithm, for fixed positive constants $B_1$ and $B_2$, and $\eta > 0$, is bounded as*

$$CovE(T) \leq T^{-\frac{1}{4}} \left( 2M + \frac{2\sqrt{2}M(1+\eta)}{\eta} + \frac{2M(1+\eta)}{\eta} B_2 B_1 (1 + o(1)) + \frac{M}{\alpha^3} \log T \right). \tag{12}$$

For the regret analysis, we consider stochastic regret, which measures the difference between the expected loss of the online algorithm and that of the best fixed miscoverage probability in hindsight. Formally, the stochastic regret is defined as:

$$\mathbb{E}[R(T)] := \sum_{t=1}^{T} \mathbb{E}[L(\bar{\alpha}_t^{\hat{m}}, \alpha_t^{\hat{m}})] - \sum_{t=1}^{T} L(\bar{\alpha}_t^{m^*}, \alpha^{m^*}) \tag{13}$$

where

$$\alpha^{m^*} = \operatorname*{arg\,min}_{\{\alpha^m, m \in [M]\}} \sum_{t=1}^{T} L(\bar{\alpha}_t^m, \alpha^m) \text{ with } \alpha^m = \operatorname*{arg\,min}_{\alpha_t^m} \sum_{t=1}^{T} L(\bar{\alpha}_t^m, \alpha_t^m), \tag{14}$$

and

$$\mathbb{E}\left[L\left(\bar{\alpha}_t^m, \alpha_t^m\right)\right] = \sum_{j=1}^{J} \bar{u}_t^j \sum_{q=1}^{\binom{M+N-1}{M}} N! \left( \prod_{m=1}^{M} \frac{(p_t^m)^{b_{m,q}}}{b_{m,q}!} \right) \sum_{m \in S_{t,q}} \bar{w}_t^{mq} L\left(\bar{\alpha}_t^m, \alpha_t^m\right). \tag{15}$$

Based on the definitions above, we establish the sublinear regret bound for the GMOCP algorithm in the following Theorem (see Appendix A.2 for detailed proof).

**Theorem 2** *GMOCP algorithm satisfies the following regret bound*

$$\mathbb{E}[R(T)] \leq \sqrt{T} \left( \frac{MT^{\frac{1}{4}}}{2\eta}(1+2\eta)^2 + \frac{\eta}{\alpha} + 2^b \ln M + T^{\frac{1}{4}}(\eta+1) + M2^{-b-1}(1+\eta)^2 \right) \tag{16}$$

---

**Algorithm 2** Graph-Structured feedback Multi-model Ensemble Online Conformal Prediction (GMOCP)

---

**Require:** $\alpha \in [0,1]$, $M$ pre-trained models, and step size $\epsilon \in (0,1)$
  **for** $t \in [T]$ **do**
    Receive new datum $x_t$.
    Generate graph $G_t$ using Algorithm 1
    Obtain $u_t^j = \sum_{m \in v_j} w_t^m, \forall j \in [J]$
    **for** $j = 1, ..., J$ **do**
      Set $p_t'^j = \frac{u_t^j}{\sum_{i=1}^{J} u_t^i}$
    **end for**
    Select one of the selective nodes according to the PMF $\boldsymbol{p}_t' = (p_t'^j)_{j=1}^{J}$.
    Create a set $S_t$ including connected models to the selected node.
    Obtain normalized weights by $\bar{w}_t^m = \frac{w_t^m}{\sum_{\bar{m} \in S_t} w_t^{\bar{m}}}, \forall m \in S_t$
    Select model $\hat{m}$ according to the PMF $\boldsymbol{w}_t^s = (w_t^{\bar{m}})_{\bar{m} \in S_t}$.
    Obtain threshold $\hat{q}_{\alpha_t^{\hat{m}}}^{\hat{m}}$ according to equation 1, and construct prediction set $C_{\alpha_t^{\hat{m}}}^{\hat{m}}(X_t)$ via equation 2.
    Observe the true label.
    Calculates $l_t^{\bar{m}}$ and update $w_t^{\bar{m}}$ and $\alpha_t^{\bar{m}}$ according to equation 8, equation 7, and equation 5 $\forall \bar{m} \in S_t$
  **end for**

---

### 3.3 Efficient GMOCP

In the previous subsection, a graph-structured feedback method was introduced to select a model $\hat{m}$ from a subset of effective models $S_t$, instead of the entire set of $M$ models. While GMOCP effectively prunes

learning models that tend to construct inefficient prediction sets, its performance can be further improved by incorporating the size of the prediction sets as an additional factor in model selection. In cases where all or most of the effective models cover the true label, the loss function may yield close values across models. In such scenarios, incorporating prediction set size helps differentiate between models by favoring those that achieve smaller sets. In this subsection, we propose EGMOCP to directly incorporate the prediction set size. Specifically, instead of relying solely on the loss-based term in the exponential update equation 7, we use a linear combination of the loss and the prediction set size to update the weights $w_t^m$ for $m \in S_t$ as:

$$w_{t+1}^m = w_t^m \exp\left(-\epsilon\left((1-\beta)\frac{l_t^m}{2^b} + \beta Len\left(\alpha_t^m\right)\mathbb{I}\{m \in S_t\}\right)\right), \tag{17}$$

where $Len(\alpha_t^m)$ is the length of the prediction set that has been created based on miscoverage probability $\alpha_t^m$, and $\beta \in (0, 1)$. The new update rule in equation 17 aims to reduce the probability of selecting models that result in large prediction sets. The following two Theorems show that the EGMOCP algorithm guarantees bounded coverage error and sublinear regret. (Proofs can be found A.3 and A.4).

**Theorem 3** *The coverage error of the EGMOCP algorithm, for fixed positive constants $C_1$ and $B_2$, and $\eta > 0$, is bounded as*

$$CovE(T) \leq T^{-\frac{1}{4}}\left(2M + \frac{2\sqrt{2}M(1+\eta)}{\eta} + \frac{2M(1+\eta)}{\eta}B_2C_1(1+o(1)) + \frac{M}{\alpha^3}\log T\right). \tag{18}$$

**Theorem 4** *EGMOCP algorithm satisfies the following regret bound*

$$\mathbb{E}[R(T)] \leq \sqrt{T}\left(\frac{MT^{\frac{1}{4}}}{2\eta}(1+2\eta)^2 + \frac{\eta}{\alpha}\right)$$
$$+ \sqrt{T}\left(\frac{\sqrt{T}}{\sqrt{T}-1}2^b\left(\ln M + N_{labels}\right) + T^{\frac{1}{4}}(\eta+1) + \frac{\sqrt{T}-1}{\sqrt{T}}\frac{M(1+\eta)^2}{2^{b+1}} + \frac{(1+\eta)N_{labels}}{\sqrt{T}} + \frac{1}{T-\sqrt{T}}2^{b-1}N_{labels}^2\right) \tag{19}$$

## 4 Experiments

This section verifies how the proposed algorithms, GMOCP and EGMOCP, result in more efficient prediction sets while covering the true label with the desired probability in practice. We first explain the experimental settings used, and then compare the performance of our two proposed methods with a multi-model conformal prediction algorithm. Note that throughout the experiments in this section, the desired miscoverage probability $\alpha$ is 0.1. All experiments were performed on a workstation with NVIDIA RTX A4000 GPU.

### 4.1 Experiental Settings

**Dataset:** We utilize corrupted versions of CIFAR-10 and CIFAR-100 (Krizhevsky et al., 2009), known as CIFAR-10C and CIFAR-100C (Hendrycks & Dietterich, 2019). These datasets consist of 15 corruption types (e.g., brightness, Gaussian noise, etc.) spanning 5 distinct levels of severity. To evaluate the effectiveness of the two proposed algorithms, we consider two distinct settings: gradual and sudden distribution shifts. In both settings, the severity of corruption changes (increases or decreases) after each batch of data. In the gradual setting, severity starts at level 0 (uncorrupted data) and increases step-by-step with each batch until it reaches level 5. It then decreases back to level 0, continuing this cycle throughout the experiment until time $T$. This setup simulates a smooth, evolving shift in the data distribution. In the sudden setting, we evaluate the algorithms' ability to handle abrupt changes. Here, the severity alternates between uncorrupted data (severity level 0) and the most severely corrupted version (severity level 5) after each batch, representing an extreme case of distribution shift. For both settings, the data sequence is split into batches of 500 data samples each. Each dataset is divided into a training phase (50,000 samples) and a test set (6,000 samples). Additionally, a separate set of 2,000 samples is used for hyperparameter selection in the conformal prediction task.

**Learning Models:** We employ 6 candidate learning models: GoogLeNet (Szegedy et al., 2015), ResNet-50, ResNet-18 He et al. (2016), DenseNet121 (Huang et al., 2017), MobileNetV2 (Sandler et al., 2018), and EfficientNet-B0 (Tan & Le, 2019). To ensure a diverse range of performance across these models, each one is trained under 3 distinct settings: High-performance setting (the model is trained for 120 epochs and initialized with default pretrained weights from ImageNet (Deng et al., 2009)), Medium-performance setting (the model is trained for only 10 epochs and initialized with random weights, resulting in weaker performance), Low-performance setting (the model is trained for just 1 epoch with random weight initialization, yielding the weakest performance among the 3). For clarity, we label each model according to its architecture and training setting. For example, the three versions of DenseNet121 are denoted as: DenseNet121-120D (120 epochs, pretrained weights), DenseNet121-10N (10 epochs, random initialization), and DenseNet121-1N (1 epoch, random initialization). In all 3 settings, the learning rate is set to $10^{-3}$, and the batch size is fixed at 64.

**Score Functions:** The nonconformity score defined in (Angelopoulos et al., 2020) is utilized to construct prediction sets. Let

$$S^m(X, Y) = \xi \sqrt{\max([k_Y - k_{reg}], 0)} + U_t \hat{f}_Y^m(X) + \rho(X, Y), \tag{20}$$

where $\hat{f}_Y^m(X)$ denotes the probability of predicting label $Y$ for input $X$ by model $m$, and $U_t$ is a random variable sampled from a uniform distribution over the interval $[0, 1]$. The term $k_Y := |\{Y' \in \mathcal{Y} \mid \hat{f}_{Y'}^m(X) \geq \hat{f}_Y^m(X)\}|$ denotes the number of labels that have a higher or equal predicted probability than label $Y$ according to the model's output probability distribution, e.g., the softmax output. $\rho(X, Y) := \sum_{Y'=1}^{N_{\text{labels}}} \hat{f}_{Y'}^m(X) \mathbb{I}[\hat{f}_{Y'}^m(X) > \hat{f}_Y^m(X)]$ sums up the probabilities of all labels that have a higher predicted probability than label $Y$. The hyperparameters $\xi$ and $k_{reg}$ are set to 0.02 and 5 for CIFAR-100C, and 0.1 and 1 for Cifar-10C, respectively.

**Evaluation Metrics:** Coverage measures the percentage of instances in which the true label is included in the prediction sets constructed by the conformal prediction algorithm over the period $[T]$. Avg Width represents the average size of the prediction sets constructed from $t = 1$ to $T$. Run Time indicates the time required to complete the algorithm for one random seed. Lastly, Single Width measures the percentage that prediction sets contain exactly one element while accurately covering the true label, highlighting cases that are most informative for predictions.

**Baseline:** The two proposed methods are compared with MOCP (Hajihashemi & Shen, 2024) and COMA (Gasparin & Ramdas, 2024a) algortihms. The MOCP algorithm employs $M$ learning models and selects one model from the entire set at each time step $t$. The selection is based on the weights assigned to each model, and the prediction set is constructed using the selected model. The COMA algorithm obtains the prediction set according to each specific learning model and then creates the final prediction set as:

$$C_{\alpha_t}(X_t) = \{Y \in \mathcal{Y} \mid \sum_{m=1}^{M} w_m^t \mathbb{I}\{Y \in C_{\alpha_t}^m(X_t)\} > \frac{1 + U(t)}{2}\}, \tag{21}$$

where $U(t)$ is a random variable uniformly distributed in $[0, 1]$, and the weights $\{w_m^t\}_{m=1}^M$ are updated over time based on the performance of each model. Specifically, the weights are inversely proportional to the exponential of the corresponding prediction set size.

Note that for all experiments conducted on CIFAR-10C and CIFAR-100C in this section, the parameters $\epsilon, \eta$, and $\beta$ were selected through grid search, with values of $0.5, 0.05$, and $0.05$, respectively. Additionally, we set $T = 6000$, indicating that the algorithm receives sequential data in an online manner over 6000 time steps.

## 4.2 Results

For this section, experiments are conducted using a candidate set of eight different learning models, including: DenseNet121-120D, ResNet-18-120D, GoogLeNet-120D, ResNet-50-120D, MobileNetV2-120D, EfficientNet-B0-120D, DenseNet121-10R, and DenseNet121-1R. We evaluate performance across various configurations, varying the maximum number of learning models connected to each selective node $N \in \{1, 3, 5\}$, and setting

the number of selective nodes $J \in \{1, 2, 4\}$. Table 1 demonstrates how, in an online setting where the data distribution experiences abrupt shifts—i.e., significant differences between two successive batches, GMOCP achieves better performance in terms of both average width and run time across all evaluated settings compared to MOCP, and has lower computational complexity compared to COMA when the number of selective nodes and model nodes is small. EGMOCP, which is specifically designed to further reduce prediction set sizes while maintaining valid coverage, achieves this goal effectively. As shown in the table, EGMOCP significantly reduces the average prediction set size and improves the single-width metric.

Table 1: Results on the CIFAR-100C dataset under sudden distribution shifts, evaluated across different values of $N$ and $J$. The target coverage is 90%. Bold numbers denote the best results in each column. GMOCP consistently achieves faster runtime compared to MOCP across all settings. EGMOCP constructs smaller prediction sets and a higher proportion of single-width sets.

| $N$ | $J$ | Method | Coverage (%) | Avg Width | Run Time | Single Width |
|---|---|---|---|---|---|---|
| | | MOCP | $89.71 \pm 0.37$ | $12.63 \pm 3.53$ | $9.37 \pm 0.05$ | $22.43 \pm 2.53$ |
| | | COMA | $\mathbf{90.00 \pm 0.01}$ | $8.36 \pm 0.95$ | $11.23 \pm 0.07$ | $28.60 \pm 1.83$ |
| 1 | 1 | GMOCP | $89.11 \pm 0.21$ | $12.03 \pm 2.80$ | $\mathbf{4.88 \pm 0.02}$ | $22.55 \pm 3.62$ |
| | | EGMOCP | $89.10 \pm 0.28$ | $6.91 \pm 0.25$ | $6.01 \pm 0.02$ | $28.62 \pm 0.93$ |
| | 2 | GMOCP | $89.10 \pm 0.19$ | $12.07 \pm 0.45$ | $4.97 \pm 0.02$ | $23.55 \pm 0.63$ |
| | | EGMOCP | $89.03 \pm 0.17$ | $7.04 \pm 0.14$ | $6.16 \pm 0.03$ | $28.36 \pm 0.91$ |
| | 4 | GMOCP | $89.04 \pm 0.21$ | $11.46 \pm 0.48$ | $5.53 \pm 0.03$ | $24.12 \pm 0.93$ |
| | | EGMOCP | $88.99 \pm 0.21$ | $6.92 \pm 0.19$ | $6.87 \pm 0.02$ | $28.50 \pm 0.80$ |
| 3 | 1 | GMOCP | $89.55 \pm 0.28$ | $10.68 \pm 1.05$ | $6.05 \pm 0.04$ | $23.45 \pm 2.53$ |
| | | EGMOCP | $89.38 \pm 0.22$ | $6.79 \pm 0.19$ | $8.80 \pm 0.12$ | $29.04 \pm 0.69$ |
| | 2 | GMOCP | $89.50 \pm 0.26$ | $10.93 \pm 0.53$ | $6.63 \pm 0.04$ | $24.49 \pm 0.63$ |
| | | EGMOCP | $89.29 \pm 0.21$ | $6.48 \pm 0.09$ | $9.30 \pm 0.09$ | $29.03 \pm 0.55$ |
| | 4 | GMOCP | $89.64 \pm 0.34$ | $10.81 \pm 0.35$ | $7.44 \pm 0.03$ | $24.30 \pm 0.64$ |
| | | EGMOCP | $89.38 \pm 0.31$ | $6.26 \pm 0.14$ | $10.29 \pm 0.03$ | $29.62 \pm 0.68$ |
| 5 | 1 | GMOCP | $89.55 \pm 0.30$ | $11.05 \pm 1.24$ | $6.91 \pm 0.07$ | $23.78 \pm 1.90$ |
| | | EGMOCP | $89.46 \pm 0.36$ | $6.59 \pm 0.12$ | $11.16 \pm 0.04$ | $29.10 \pm 0.65$ |
| | 2 | GMOCP | $89.53 \pm 0.26$ | $11.35 \pm 1.17$ | $7.87 \pm 0.06$ | $23.85 \pm 0.95$ |
| | | EGMOCP | $89.44 \pm 0.28$ | $6.30 \pm 0.13$ | $12.03 \pm 0.35$ | $29.81 \pm 0.46$ |
| | 4 | GMOCP | $89.73 \pm 0.31$ | $11.65 \pm 0.81$ | $8.94 \pm 0.07$ | $23.62 \pm 0.62$ |
| | | EGMOCP | $89.43 \pm 0.27$ | $\mathbf{6.18 \pm 0.14}$ | $12.97 \pm 0.05$ | $\mathbf{29.91 \pm 0.47}$ |

To enable selective nodes with different levels of exploration and exploitation—as considered in (Ghari & Shen, 2020)—we use different $\eta_e$ values across selective nodes in equation 10. For the case $J = 2$, we use $\eta_e = \{0.2, 0.8\}$, and for $J = 4$, we use $\eta_e = \{0.1, 0.2, 0.3, 0.4\}$. This setup ensures that the last selective node places more emphasis on exploration compared to the first node. Table 2 presents experimental results on CIFAR-10C under gradual distribution shifts. Note that across all settings, GMOCP and EGMOCP algorithms achieve coverage close to the desired level of 90%. GMOCP consistently demonstrates lower computational complexity compared to MOCP while producing smaller prediction sets and a higher proportion of single-width sets. GMOCP is also faster than COMA in cases where there are a small number of selective nodes and model nodes. Furthermore, EGMOCP consistently constructs significantly smaller prediction sets and yields a higher proportion of single-width prediction sets compared to all benchmarks. Note that even though COMA obtains the desired coverage experimentally, such coverage is not guaranteed theoretically.

To demonstrate the effect of using prediction set size as feedback—which enables EGMOCP to constructs more efficient prediction sets compared to GMOCP and MOCP—Figure 2 illustrates the behavior of EGMOCP across the entire time horizon. To better visualize the differences among the 3 proposed training configurations, the figure shows results for three versions of DenseNet121 under the setting $N = 5$ and $J = 2$. As observed, better-performing models tend to construct smaller prediction sets. By incorporating prediction set size into the update rule equation 17, the algorithm assigns lower weights $w_t^m$ to weaker models, reducing their chances of being selected. As a result, EGMOCP favors high-performing models and constructs significantly smaller prediction sets than the other two algorithms.

Table 2: Results on the CIFAR-10C dataset under gradual distribution shifts, evaluated across different values of $N$ and $J$. The target coverage is 90%. Bold numbers denote the best results in each column. GMOCP consistently achieves faster runtime compared to MOCP across all settings. EGMOCP constructs smaller prediction sets and a higher proportion of single-width sets.

| $N$ | $J$ | Method | Coverage (%) | Avg Width | Run Time | Single Width |
|---|---|---|---|---|---|---|
| | | MOCP | **90.03** $\pm$ 0.30 | 2.07 $\pm$ 0.35 | 8.83 $\pm$ 0.04 | 48.00 $\pm$ 7.84 |
| | | COMA | 90.02 $\pm$ 0.02 | 1.49 $\pm$ 0.07 | 10.76 $\pm$ 0.04 | 61.39 $\pm$ 2.92 |
| 1 | 1 | GMOCP | 89.36 $\pm$ 0.21 | 1.90 $\pm$ 0.27 | **4.32** $\pm$ **0.01** | 48.14 $\pm$ 4.22 |
| | | EGMOCP | 89.37 $\pm$ 0.22 | 1.52 $\pm$ 0.03 | 5.51 $\pm$ 0.06 | 57.48 $\pm$ 1.58 |
| | 2 | GMOCP | 89.37 $\pm$ 0.17 | 1.78 $\pm$ 0.03 | 4.39 $\pm$ 0.03 | 52.30 $\pm$ 1.27 |
| | | EGMOCP | 89.35 $\pm$ 0.21 | 1.59 $\pm$ 0.02 | 5.58 $\pm$ 0.01 | 55.57 $\pm$ 1.19 |
| | 4 | GMOCP | 89.25 $\pm$ 0.21 | 1.77 $\pm$ 0.04 | 4.74 $\pm$ 0.03 | 52.45 $\pm$ 1.03 |
| | | EGMOCP | 89.28 $\pm$ 0.16 | 1.55 $\pm$ 0.02 | 5.91 $\pm$ 0.01 | 56.69 $\pm$ 1.11 |
| 3 | 1 | GMOCP | 89.79 $\pm$ 0.25 | 1.78 $\pm$ 0.17 | 5.41 $\pm$ 0.06 | 50.97 $\pm$ 3.41 |
| | | EGMOCP | 89.83 $\pm$ 0.30 | 1.50 $\pm$ 0.02 | 8.44 $\pm$ 0.03 | 58.98 $\pm$ 1.07 |
| | 2 | GMOCP | 89.78 $\pm$ 0.32 | 1.78 $\pm$ 0.04 | 6.06 $\pm$ 0.05 | 52.64 $\pm$ 1.23 |
| | | EGMOCP | 89.65 $\pm$ 0.22 | 1.37 $\pm$ 0.01 | 9.08 $\pm$ 0.05 | 61.51 $\pm$ 0.77 |
| | 4 | GMOCP | 89.72 $\pm$ 0.28 | 1.73 $\pm$ 0.03 | 6.36 $\pm$ 0.04 | 53.68 $\pm$ 1.17 |
| | | EGMOCP | 89.60 $\pm$ 0.35 | 1.41 $\pm$ 0.02 | 9.73 $\pm$ 0.02 | 60.68 $\pm$ 1.21 |
| 5 | 1 | GMOCP | 89.72 $\pm$ 0.17 | 1.85 $\pm$ 0.15 | 6.71 $\pm$ 0.11 | 51.02 $\pm$ 2.96 |
| | | EGMOCP | 89.82 $\pm$ 0.26 | 1.48 $\pm$ 0.02 | 10.88 $\pm$ 0.04 | 59.39 $\pm$ 0.83 |
| | 2 | GMOCP | 89.73 $\pm$ 0.26 | 1.80 $\pm$ 0.03 | 7.33 $\pm$ 0.03 | 52.27 $\pm$ 1.13 |
| | | EGMOCP | 89.87 $\pm$ 0.36 | **1.35** $\pm$ **0.01** | 11.66 $\pm$ 0.03 | **62.23** $\pm$ **0.57** |
| | 4 | GMOCP | 89.88 $\pm$ 0.21 | 1.80 $\pm$ 0.02 | 8.47 $\pm$ 0.07 | 52.49 $\pm$ 0.92 |
| | | EGMOCP | 89.99 $\pm$ 0.24 | 1.38 $\pm$ 0.02 | 12.84 $\pm$ 0.11 | 61.84 $\pm$ 0.55 |

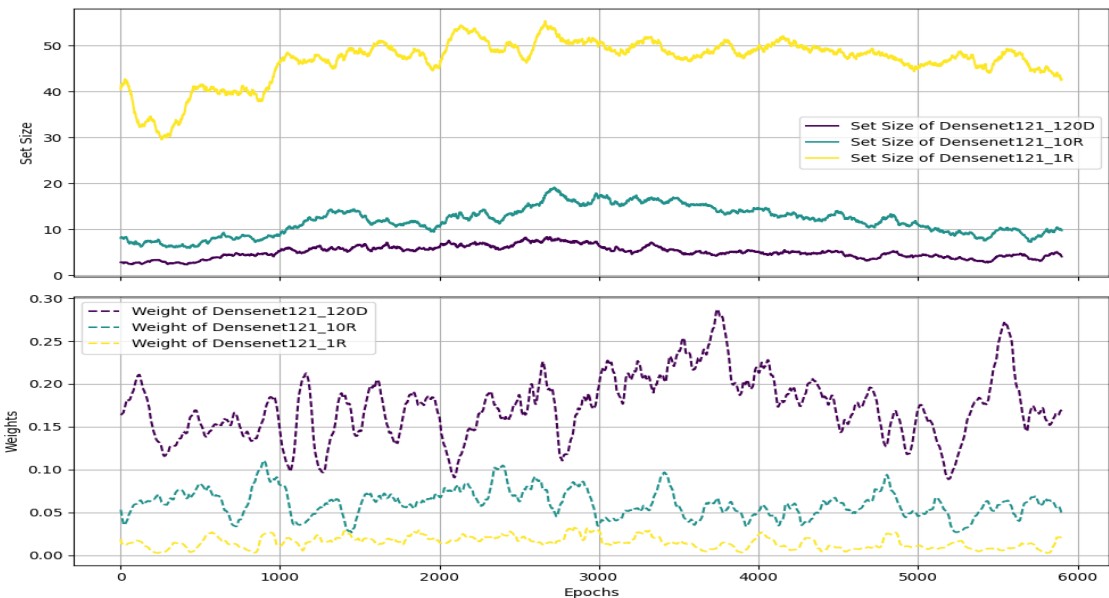

Figure 2: Evaluation of prediction sets constructed by 3 training configurations of DenseNet121 under $N = 5$ and $J = 2$ over 6000 timesteps. The top plot shows the size of the prediction sets, while the bottom plot shows the corresponding model weights $w_t^m$ over time. DenseNet121-120D consistently receives the highest weight, indicating a higher likelihood of being selected. Moreover, models with better performance (e.g., DenseNet121-120D) create significantly smaller prediction sets.

## 5 Conclusion

In this paper, we proposed 2 multi-model online conformal prediction algorithms. By leveraging graph-based feedback, the proposed methods dynamically select a subset of effective learning models at each time step for the conformal prediction task. Additionally, we demonstrated that incorporating prediction set

size as feedback—alongside the loss function—into the model weight update rule significantly improves the efficiency of the constructed prediction sets. This results in smaller prediction sets while still satisfying the required coverage guarantee. It is proved theoretically that the 2 proposed algorithms guarantee valid coverage and achieve sublinear regret. Experimental results in an online setting, simulating real-world online environments, show that GMOCP consistently creates smaller prediction sets with lower computational complexity compared to baselines. Furthermore, EGMOCP is able to construct even smaller prediction sets by effectively selecting high-performance models.

## 6 Acknowledgments

Work in the paper is supported by NSF EECS 2207457, NSF ECCS 2412484 and NSF ECCS 2442964.

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

# A Proofs

## A.1 Proof of Theorem 1

Define $a_t^m := \alpha - err_t^m$. Suppose the sequence $\{a_t^m\}_{t \in [T]} \in \mathbb{R}$ and satisfies $\alpha \le |a_t^m| \le 1$ for $0 < \alpha < 1$. The proof of this lemma is based on a grouping argument. We start by defining new variables $L$ and $K$ as follows

$$L = \lceil T^\gamma \rceil, \quad K = \lceil \frac{T}{L} \rceil \le T^{1-\gamma} + 1,$$

where $\gamma \in (0, 1)$ is a parameter to be chosen. The $k$th group out of all $K$ groups is defined by

$$G_k = \{t_{k-1} + 1, \dots, t_k\} := \{(k-1)L + 1, \dots, \min(kL, T)\}.$$

This results in $\bigcup_{k=1}^{K} G_k = [T]$, with $|G_k| = L$ for all $k \in [K-1]$, and $|G_K| \le L$. For any $k \ge 2$, we define

$$S_k := \sum_{t \in G_k} \sum_{q=1}^{Q} N! \left( \prod_{m=1}^{M} \frac{(p_t^m)^{b_{m,q}}}{b_{m,q}!} \right) \sum_{m \in S_{t,q}} \frac{\bar{w}_t^{mq} a_t^m}{\sqrt{\sum_{\tau=1}^{t} (a_\tau^m)^2}}$$

$$\bar{S}_k := \sum_{t \in G_k} \sum_{q=1}^{Q} N! \left( \prod_{m=1}^{M} \frac{(p_t^m)^{b_{m,q}}}{b_{m,q}!} \right) \sum_{m \in S_{t,q}} \frac{\bar{w}_t^{mq} a_t^m}{\sqrt{\sum_{\tau=1}^{t_k - 1} (a_\tau^m)^2}}, \tag{22}$$

For $S_k$ and $\bar{S}_k$ we have

$$|S_k - \bar{S}_k|$$

$$\le \sum_{t \in G_k} \sum_{q=1}^{Q} N! \left( \prod_{m=1}^{M} \frac{(p_t^m)^{b_{m,q}}}{b_{m,q}!} \right) \sum_{m \in S_{t,q}} \bar{w}_t^{mq} |a_t^m| \left( \frac{1}{\sqrt{\sum_{\tau=1}^{t_k-1} (a_\tau^m)^2}} - \frac{1}{\sqrt{\sum_{\tau=1}^{t} (a_\tau^m)^2}} \right)$$

$$\le |G_k| \sum_{m=1}^{M} \left( \frac{1}{\sqrt{\sum_{\tau=1}^{t_k-1} (a_\tau^m)^2}} - \frac{1}{\sqrt{\sum_{\tau=1}^{t_k} (a_\tau^m)^2}} \right) \overset{(i)}{\le} L \sum_{m=1}^{M} \left( \frac{\sum_{\tau=t_{k-1}+1}^{t_k} (a_\tau^m)^2}{2 \left( \sum_{\tau=1}^{t_k-1} (a_\tau^m)^2 \right)^{\frac{3}{2}}} \right)$$

$$\overset{(ii)}{\le} L \sum_{m=1}^{M} \frac{L}{2\alpha^2(k-1)L\sqrt{\sum_{\tau=1}^{t_k-1}(a_\tau^m)^2}} \le \frac{L}{2\alpha^2(k-1)} \sum_{m=1}^{M} \frac{1}{\sqrt{\alpha^2(k-1)L}}$$

$$= \frac{ML}{2\alpha^3(k-1)\sqrt{(k-1)L}}, \tag{23}$$

where $(i)$ employed inequality $\left( \frac{1}{\sqrt{x}} - \frac{1}{\sqrt{x+y}} \right) \le \frac{y}{2x^{\frac{3}{2}}}$ for $x, y \ge 0$. Additionally $(ii)$ considered two bounds $\sum_{t_{k-1}+1}^{t_k} (a_\tau^m)^2 \le (t_k - t_{k-1}) \le L$ and $\sum_{\tau=1}^{t_k-1} (a_\tau^m)^2 \ge \alpha^2 t_{k-1} = \alpha^2(k-1)L$. By using triangle inequality we have:

$$|\bar{S}_k| \le |S_k| + |\bar{S}_k - S_k| \le |S_k| + \frac{ML}{2\alpha^3(k-1)\sqrt{(k-1)L}}.$$

For any $k \geq 2$ we have

$$
\begin{aligned}
&\left| \sum_{t \in G_k} \sum_{q=1}^{Q} N! \left( \prod_{m=1}^{M} \frac{(p_t^m)^{b_{m,q}}}{b_{m,q}!} \right) \sum_{m \in S_{t,q}} \bar{w}_t^{mq} a_t^m \right| \\
&= \left| \sum_{t \in G_k} \sum_{q=1}^{Q} N! \left( \prod_{m=1}^{M} \frac{(p_t^m)^{b_{m,q}}}{b_{m,q}!} \right) \sum_{m \in S_{t,q}} \bar{w}_t^{mq} a_t^m \frac{\sqrt{\sum_{\tau=1}^{t_k-1}(a_\tau^m)^2}}{\sqrt{\sum_{\tau=1}^{t_k-1}(a_\tau^m)^2}} \right| \\
&\leq \left| \sum_{t \in G_k} \sum_{q=1}^{Q} N! \left( \prod_{m=1}^{M} \frac{(p_t^m)^{b_{m,q}}}{b_{m,q}!} \right) \sum_{m \in S_{t,q}} \bar{w}_t^{mq} a_t^m \frac{1}{\sqrt{\sum_{\tau=1}^{t_k-1}(a_\tau^m)^2}} \right| \cdot \sqrt{(k-1)L} \\
&\leq (|S_k| + |\bar{S}_k - S_k|) \cdot \sqrt{(k-1)L} \leq |S_k|\sqrt{(k-1)L} + \frac{ML}{2\alpha^3(k-1)}.
\end{aligned}
\tag{24}
$$

According to equation 5 and defining $A_t^m := \sum_{q=1}^{Q} N! \left( \prod_{m=1}^{M} \frac{(p_t^m)^{b_{m,q}}}{b_{m,q}!} \right) \bar{w}_t^{mq}$, for $|S_k|$ we have:

$$
\begin{aligned}
|S_k| &= \left| \sum_{t \in G_k} \sum_{q=1}^{Q} N! \left( \prod_{m=1}^{M} \frac{(p_t^m)^{b_{m,q}}}{b_{m,q}!} \right) \sum_{m \in S_{t,q}} \frac{\bar{w}_t^{mq}(\alpha_{t+1}^m - \alpha_t^m)}{\eta} \right| = \frac{1}{\eta} \left| \sum_{m=1}^{M} \sum_{t \in G_k} A_t^m(\alpha_{t+1}^m - \alpha_t^m) \right| \\
&\leq \frac{1}{\eta} \sum_{m=1}^{M} \left| -A_{t_{k-1}+1}^m \alpha_{t_{k-1}+1}^m + A_{t_k}^m \alpha_{t_k+1}^m + \alpha_{t_{k-1}+2}^m(A_{t_{k-1}+1}^m - A_{t_{k-1}+2}^m) + ... + \alpha_{t_k}^m(A_{t_k-1}^m - A_{t_k}^m) \right| \\
&\overset{(i)}{\leq} \frac{1}{\eta} \sum_{m=1}^{M} 2(1+\eta) + |G_k|(1+\eta)B_2\left(e^{B_1\epsilon} - 1\right) = \frac{M(1+\eta)}{\eta}\left(2 + |G_k|B_2\left(e^{B_1\epsilon} - 1\right)\right),
\end{aligned}
\tag{25}
$$

where $(i)$ follows the upper bound obtained in Lemma 3. Also for $k = 1$ we have $|\sum_{t \in G_1} \sum_{q=1}^{\binom{M+N-1}{M}} N! \left( \prod_{m=1}^{M} \frac{(p_t^m)^{b_{m,q}}}{b_{m,q}!} \right) \sum_{m \in S_{t,q}} \bar{w}_t^{mq} a_t^m| \leq M|G1| \leq ML$. By summing bounds over $k \in [K]$ we have

$$
\begin{aligned}
&\left| \sum_{t=1}^{T} \sum_{q=1}^{Q} N! \left( \prod_{m=1}^{M} \frac{(p_t^m)^{b_{m,q}}}{b_{m,q}!} \right) \sum_{m \in S_{t,q}} \bar{w}_t^{mq} a_t^m \right| \\
&\leq ML + \sum_{k=2}^{K} \left| \sum_{t \in G_k} \sum_{q=1}^{Q} N! \left( \prod_{m=1}^{M} \frac{(p_t^m)^{b_{m,q}}}{b_{m,q}!} \right) \sum_{m \in S_{t,q}} \bar{w}_t^{mq} a_t^m \right| \\
&\leq ML + \sum_{k=2}^{K} \frac{M(1+\eta)}{\eta}\left(2 + LB_2\left(e^{B_1\epsilon} - 1\right)\right)\sqrt{(k-1)L} + \frac{ML}{2\alpha^3} \sum_{k=2}^{K} \frac{1}{k-1} \\
&\leq ML + \frac{2M(1+\eta)}{\eta}\left(\sqrt{L}K^{\frac{3}{2}} + (KL)^{\frac{3}{2}}B_2(e^{B_1\epsilon} - 1)\right) + \frac{ML}{2\alpha^3}\log K \\
&\leq M\lceil T^\gamma \rceil + \frac{2M(1+\eta)}{\eta}\left(\sqrt{2}T^{\frac{3}{2}-\gamma} + T^{\frac{3}{2}}B_2(e^{B_1\epsilon} - 1)\right) + \frac{M\lceil T^\gamma \rceil}{2\alpha^3}\log T^{1-\gamma} \\
&\overset{(i)}{\leq} 2MT^\gamma + \frac{2M(1+\eta)}{\eta}\left(\sqrt{2}T^{\frac{3}{2}-\gamma} + T^{\frac{3}{2}}B_2(B_1\epsilon + \frac{(B_1\epsilon)^2}{2!} + ...)\right) + \frac{MT^\gamma}{\alpha^3}\log T.
\end{aligned}
\tag{26}
$$

The exponential term is replaced with its Taylor series in $(i)$. We can achieve convergence for coverage error by setting $\gamma \in \left(\frac{1}{2}, 1\right)$ and $\epsilon = T^{-\frac{3}{4}}$. Choosing $\gamma = \frac{3}{4}$ and dividing by $T$, we obtain

$$\left| \frac{1}{T} \sum_{t=1}^{T} \sum_{q=1}^{Q=\binom{M+N-1}{M}} N! \left( \prod_{m=1}^{M} \frac{(p_t^m)^{b_{m,q}}}{b_{m,q}!} \right) \sum_{m \in S_{t,q}} \bar{w}_t^{mq} a_t^m \right|$$

$$\leq T^{-\frac{1}{4}} \left( 2M + \frac{2\sqrt{2}M(1+\eta)}{\eta} + \frac{2M(1+\eta)}{\eta} B_2 B_1 (1 + \frac{B_1 T^{-\frac{1}{4}}}{2!} + ...) + \frac{M}{\alpha^3} \log T \right).$$

$$= T^{-\frac{1}{4}} \left( 2M + \frac{2\sqrt{2}M(1+\eta)}{\eta} + \frac{2M(1+\eta)}{\eta} B_2 B_1 (1 + o(1)) + \frac{M}{\alpha^3} \log T \right). \tag{27}$$

## A.2 Proof of Theorem 2

To prove regret bound for GMOCP, we first establish the following two lemmas as a stepstone.

**Lemma 1** *For miscoverage probability assigned to any model $\tilde{m} \in [M]$, we have the following bound*

$$\sum_{t=1}^{T} L(\bar{\alpha}_t^{\tilde{m}}, \alpha_t^{\tilde{m}}) - \sum_{t=1}^{T} L(\bar{\alpha}_t^{\tilde{m}}, \alpha^{\tilde{m}}) \leq \frac{M\sqrt{T}}{2\eta \eta_e}(1 + 2\eta)^2 + \frac{\eta\sqrt{T}}{\alpha},$$

*where $\alpha^{\tilde{m}} = \arg\min_{\alpha_t^{\tilde{m}}} \sum_{t=1}^{T} L(\bar{\alpha}_t^{\tilde{m}}, \alpha_t^{\tilde{m}})$.*
**Proof:** *We first begin with*

$$(\alpha_{t+1}^{\tilde{m}} - \alpha^{\tilde{m}})^2 = (\alpha_t^{\tilde{m}} - \eta \frac{\nabla_{\alpha_t^{\tilde{m}}} L(\bar{\alpha}_t^{\tilde{m}}, \alpha_t^{\tilde{m}}) \mathbb{I}\{\tilde{m} \in S_t\}}{\sqrt{\sum_{\tau=1}^{t} \|\nabla_{\alpha_\tau^{\tilde{m}}} L(\bar{\alpha}_\tau^{\tilde{m}}, \alpha_\tau^{\tilde{m}})\|_2^2}} - \alpha^{\tilde{m}})^2.$$

*Then define adaptive learning rate $\eta_t$ as*

$$\eta_t := \frac{\eta}{\sqrt{\sum_{\tau=1}^{t} \|\nabla_{\alpha_\tau^{\tilde{m}}} L(\bar{\alpha}_\tau^{\tilde{m}}, \alpha_\tau^{\tilde{m}})\|_2^2}},$$

*where $\frac{\eta}{\sqrt{t}} \leq \eta_t \leq \frac{\eta}{\alpha\sqrt{t}}$. Then we have*

$$(\alpha_{t+1}^{\tilde{m}} - \alpha^{\tilde{m}})^2 = (\eta_t \nabla_{\alpha_t^{\tilde{m}}} L(\bar{\alpha}_t^{\tilde{m}}, \alpha_t^{\tilde{m}}) \mathbb{I}\{\tilde{m} \in S_t\})^2 + (\alpha_t^{\tilde{m}} - \alpha^{\tilde{m}})^2 - 2\eta_t(\alpha_t^{\tilde{m}} - \alpha^{\tilde{m}})\nabla_{\alpha_t^{\tilde{m}}} L(\bar{\alpha}_t^{\tilde{m}}, \alpha_t^{\tilde{m}})\mathbb{I}\{\tilde{m} \in S_t\}.$$

*Therefore,*

$$(\alpha_t^{\tilde{m}} - \alpha^{\tilde{m}})\nabla_{\alpha_t^{\tilde{m}}} L(\bar{\alpha}_t^{\tilde{m}}, \alpha_t^{\tilde{m}})\mathbb{I}\{\tilde{m} \in S_t\} = \frac{(\alpha_t^{\tilde{m}} - \alpha^{\tilde{m}})^2 - (\alpha_{t+1}^{\tilde{m}} - \alpha^{\tilde{m}})^2}{2\eta_t} + \frac{\eta_t}{2}(\nabla_{\alpha_t^{\tilde{m}}} L(\bar{\alpha}_t^{\tilde{m}}, \alpha_t^{\tilde{m}})\mathbb{I}\{\tilde{m} \in S_t\})^2. \tag{28}$$

*Since the loss function equation 3 is convex, we have the following inequality*

$$L(\bar{\alpha}_t^{\tilde{m}}, \alpha_t^{\tilde{m}}) - L(\bar{\alpha}_t^{\tilde{m}}, \alpha^{\tilde{m}}) \leq (\alpha_t^{\tilde{m}} - \alpha^{\tilde{m}})\nabla_{\alpha_t^{\tilde{m}}} L(\bar{\alpha}_t^{\tilde{m}}, \alpha_t^{\tilde{m}}). \tag{29}$$

*Combining equation 28 and equation 29, we arrive at*

$$\left( L(\bar{\alpha}_t^{\tilde{m}}, \alpha_t^{\tilde{m}}) - L(\bar{\alpha}_t^{\tilde{m}}, \alpha^{\tilde{m}}) \right) \mathbb{I}\{\tilde{m} \in S_t\}$$
$$\leq \frac{(\alpha_t^{\tilde{m}} - \alpha^{\tilde{m}})^2 - (\alpha_{t+1}^{\tilde{m}} - \alpha^{\tilde{m}})^2}{2\eta_t} + \frac{\eta_t}{2}(\nabla_{\alpha_t^{\tilde{m}}} L(\bar{\alpha}_t^{\tilde{m}}, \alpha_t^{\tilde{m}})\mathbb{I}\{\tilde{m} \in S_t\})^2. \tag{30}$$

*Taking the expectation of left hand side of equation 30 with respect to $\mathbb{I}\{\tilde{m} \in S_t\}$, we obtain*

$$\mathbb{E}\left[ \left( L(\bar{\alpha}_t^{\tilde{m}}, \alpha_t^{\tilde{m}}) - L(\bar{\alpha}_t^{\tilde{m}}, \alpha^{\tilde{m}}) \right) \mathbb{I}\{\tilde{m} \in S_t\} \right]$$
$$= \left( L(\bar{\alpha}_t^{\tilde{m}}, \alpha_t^{\tilde{m}}) - L(\bar{\alpha}_t^{\tilde{m}}, \alpha^{\tilde{m}}) \right) \times 1 \ \times q_t^{\tilde{m}} + \left( L(\bar{\alpha}_t^{\tilde{m}}, \alpha_t^{\tilde{m}}) - L(\bar{\alpha}_t^{\tilde{m}}, \alpha^{\tilde{m}}) \right) \times 0 \ \times \left( 1 - q_t^{\tilde{m}} \right)$$
$$= q_t^m \left( L(\bar{\alpha}_t^{\tilde{m}}, \alpha_t^{\tilde{m}}) - L(\bar{\alpha}_t^{\tilde{m}}, \alpha^{\tilde{m}}) \right) \tag{31}$$

where $q_t^{\tilde{m}}$ is the probability that model $\tilde{m}$ is in the chosen subset of models. Moreover, for the expectation of right hand side of equation 30, we have

$$\mathbb{E}\left[\frac{(\alpha_t^{\tilde{m}} - \alpha^{\tilde{m}})^2 - (\alpha_{t+1}^{\tilde{m}} - \alpha^{\tilde{m}})^2}{2\eta_t} + \frac{\eta_t}{2}(\nabla_{\alpha_t^{\tilde{m}}} L(\bar{\alpha}_t^{\tilde{m}}, \alpha_t^{\tilde{m}})\mathbb{I}\{\tilde{m} \in S_t\})^2\right]$$

$$= \frac{(\alpha_t^{\tilde{m}} - \alpha^{\tilde{m}})^2 - (\alpha_{t+1}^{\tilde{m}} - \alpha^{\tilde{m}})^2}{2\eta_t} + \frac{\eta_t q_t^{\tilde{m}}}{2}(\nabla_{\alpha_t^{\tilde{m}}} L(\bar{\alpha}_t^{\tilde{m}}, \alpha_t^{\tilde{m}}))^2. \tag{32}$$

Based on equation 8, In this setting we obtain

$$q_t^m = \sum_{j=1}^{J} p'^j_t \left(1 - (1 - p_t^m)^N\right) = \sum_{j=1}^{J} p'^j_t p_t^m \left(1 + ... + (1 - p_t^m)^{N-1}\right)$$

$$\geq \sum_{j=1}^{J} p'^j_t p_t^m \geq \sum_{j=1}^{J} \bar{u}_t^j \frac{\eta_e}{M} = \frac{\eta_e}{M} \tag{33}$$

From equation 30, equation 31, and equation 32, we can conclude that

$$L(\bar{\alpha}_t^{\tilde{m}}, \alpha_t^{\tilde{m}}) - L(\bar{\alpha}_t^{\tilde{m}}, \alpha^{\tilde{m}}) \leq \frac{(\alpha_t^{\tilde{m}} - \alpha^{\tilde{m}})^2 - (\alpha_{t+1}^{\tilde{m}} - \alpha^{\tilde{m}})^2}{2\eta_t q_t^{\tilde{m}}} + \frac{\eta_t}{2}(\nabla_{\alpha_t^{\tilde{m}}} L(\bar{\alpha}_t^{\tilde{m}}, \alpha_t^{\tilde{m}}))^2. \tag{34}$$

By summing equation 34 over $t = 1, ..., T$, we have

$$\sum_{t=1}^{T} \left(L\left(\bar{\alpha}_t^{\tilde{m}}, \alpha_t^{\tilde{m}}\right) - L\left(\bar{\alpha}_t^{\tilde{m}}, \alpha^{\tilde{m}}\right)\right)$$

$$\leq \sum_{t=1}^{T} \frac{(\alpha_t^{\tilde{m}} - \alpha^{\tilde{m}})^2 - (\alpha_{t+1}^{\tilde{m}} - \alpha^{\tilde{m}})^2}{2\eta_t q_t^{\tilde{m}}} + \sum_{t=1}^{T} \frac{\eta_t}{2}\left(\nabla_{\alpha_t^{\tilde{m}}} L(\bar{\alpha}_t^{\tilde{m}}, \alpha_t^{\tilde{m}})\right)^2$$

$$\overset{(i)}{\leq} \frac{M\sqrt{T}}{2\eta} \sum_{t=1}^{T} \frac{(\alpha_t^{\tilde{m}} - \alpha^{\tilde{m}})^2 - (\alpha_{t+1}^{\tilde{m}} - \alpha^{\tilde{m}})^2}{\eta_e} + \frac{\eta}{2} \sum_{t=1}^{T} \frac{1}{\sqrt{\sum_{\tau=1}^{t} \|\nabla_{\alpha_\tau^{\tilde{m}}} L(\bar{\alpha}_\tau^{\tilde{m}}, \alpha_\tau^{\tilde{m}})\|_2^2}}$$

$$\leq \frac{M\sqrt{T}}{2\eta\eta_e} \sum_{t=1}^{T} (\alpha_t^{\tilde{m}} - \alpha^{\tilde{m}})^2 - (\alpha_{t+1}^{\tilde{m}} - \alpha^{\tilde{m}})^2 + \frac{\eta}{2} \sum_{t=1}^{T} \frac{1}{\sqrt{\sum_{\tau=1}^{t} \|\nabla_{\alpha_\tau^{\tilde{m}}} L(\bar{\alpha}_\tau^{\tilde{m}}, \alpha_\tau^{\tilde{m}})\|_2^2}}$$

$$\leq \frac{M\sqrt{T}}{2\eta\eta_e} \left((\alpha_1^{\tilde{m}} - \alpha^{\tilde{m}})^2 - (\alpha_{T+1}^{\tilde{m}} - \alpha^{\tilde{m}})^2\right) + \frac{\eta}{2} \sum_{t=1}^{T} \frac{1}{\alpha\sqrt{t}}$$

$$\overset{(ii)}{\leq} \frac{M\sqrt{T}}{2\eta\eta_e}(1 + 2\eta)^2 + \frac{\eta\sqrt{T}}{\alpha}, \tag{35}$$

where in (i) we use $q_t^m \geq \frac{\eta_e}{M}$ as in equation 33 and (ii) used $\alpha_t^m \in [-\eta, 1 + \eta]$ as proved by Lemma 2 in (Hajihashemi & Shen, 2024).

**Lemma 2** For any model $\tilde{m} \in [M]$ following bound holds

$$\sum_{t=1}^{T} \mathbb{E}\left[L\left(\bar{\alpha}_\tau^m, \alpha_\tau^m\right)\right] - \sum_{t=1}^{T} L\left(\bar{\alpha}_\tau^{\tilde{m}}, \alpha_\tau^{\tilde{m}}\right) \leq \frac{2^b}{\epsilon} \ln M + T\eta_e(\eta + 1) + T\epsilon 2^{b-1} M \frac{(1+\eta)^2}{2^{2b}} \tag{36}$$

**Proof:** Defining $W_t := \sum_{m=1}^{M} w_t^m$ and $\bar{u}_t^j := \frac{u_t^j}{U_t}$, we have

$$\frac{W_{t+1}}{W_t} = \sum_{j=1}^{J} \bar{u}_t^j \frac{W_{t+1}}{W_t} = \sum_{j=1}^{J} \bar{u}_t^j \sum_{m=1}^{M} \frac{w_{t+1}^m}{W_t} = \sum_{j=1}^{J} \bar{u}_t^j \sum_{m=1}^{M} \frac{w_t^m}{W_t} \exp\left(-\epsilon \frac{l_t^m}{2^b}\right)$$

$$= \sum_{j=1}^{J} \bar{u}_t^j \sum_{m=1}^{M} \frac{p_t^m - \frac{\eta_e}{M}}{1 - \eta_e} \exp\left(-\epsilon \frac{l_t^m}{2^b}\right) \tag{37}$$

*Using the inequality $\exp(-x) \leq 1 - x + \frac{x^2}{2}, \forall x \geq 0$ leads to*

$$\frac{W_{t+1}}{W_t} \leq \sum_{j=1}^{J} \bar{u}_t^j \sum_{m=1}^{M} \frac{p_t^m - \frac{\eta_e}{M}}{1 - \eta_e} \left( 1 - \epsilon \frac{l_t^m}{2^b} + \frac{\epsilon^2 \left(\frac{l_t^m}{2^b}\right)^2}{2} \right)$$

*By taking the logarithm from both sides of above inequality we have*

$$\ln \frac{W_{t+1}}{W_t} \leq \ln \sum_{j=1}^{J} \bar{u}_t^j \sum_{m=1}^{M} \frac{p_t^m - \frac{\eta_e}{M}}{1 - \eta_e} \left( 1 - \epsilon \frac{l_t^m}{2^b} + \frac{\epsilon^2 \left(\frac{l_t^m}{2^b}\right)^2}{2} \right)$$

$$\overset{(i)}{\leq} \sum_{j=1}^{J} \bar{u}_t^j \sum_{m=1}^{M} \frac{p_t^m - \frac{\eta_e}{M}}{1 - \eta_e} \left( 1 - \epsilon \frac{l_t^m}{2^b} + \frac{\epsilon^2 \left(\frac{l_t^m}{2^b}\right)^2}{2} \right) - 1$$

$$\overset{(ii)}{=} \sum_{j=1}^{J} \bar{u}_t^j \sum_{m=1}^{M} \frac{p_t^m - \frac{\eta_e}{M}}{1 - \eta_e} \left( -\epsilon \frac{l_t^m}{2^b} + \frac{\epsilon^2 \left(\frac{l_t^m}{2^b}\right)^2}{2} \right) \tag{38}$$

*where $\overset{(i)}{\leq}$ follows $1 + x \leq \exp x$ in case we replace $x$ with $\ln y$ which leads to $1 + \ln y \leq y$, and $\overset{(ii)}{=}$ follows $\sum_{j=1}^{J} \bar{u}_t^j \sum_{m=1}^{M} \frac{p_t^m - \frac{\eta_e}{M}}{1 - \eta_e} = 1$. Summing equation 38 over $t$ from 1 to $T$ result in*

$$\ln \frac{W_{T+1}}{W_1} \leq \sum_{t=1}^{T} \sum_{j=1}^{J} \bar{u}_t^j \sum_{m=1}^{M} \frac{p_t^m - \frac{\eta_e}{M}}{1 - \eta_e} \left( -\epsilon \frac{l_t^m}{2^b} + \frac{\epsilon^2 \left(\frac{l_t^m}{2^b}\right)^2}{2} \right) \tag{39}$$

*Furthermore, recall the updating rule of $w_t^m$ in equation 7, for any model $\tilde{m} \in [M]$ we have*

$$\ln \frac{W_{T+1}}{W_1} \geq \ln \frac{w_{T+1}^{\tilde{m}}}{W_1} = \ln w_1^{\tilde{m}} \exp \left( \sum_{t=1}^{T} -\epsilon \frac{l_t^{\tilde{m}}}{2^b} \right) - \ln 1 = -\ln M - \sum_{t=1}^{T} \epsilon \frac{l_t^{\tilde{m}}}{2^b} \tag{40}$$

*combining equation 39 with equation 40 result in*

$$\sum_{t=1}^{T} \sum_{j=1}^{J} \bar{u}_t^j \sum_{m=1}^{M} \frac{p_t^m - \frac{\eta_e}{M}}{1 - \eta_e} \left( -\epsilon \frac{l_t^m}{2^b} + \frac{\epsilon^2 \left(\frac{l_t^m}{2^b}\right)^2}{2} \right) \geq -\ln M - \sum_{t=1}^{T} \epsilon \frac{l_t^{\tilde{m}}}{2^b}$$

*Multiplying $\frac{2^b(1-\eta_e)}{\epsilon}$ to both sides and rearrangement leads to*

$$\sum_{t=1}^{T} \sum_{j=1}^{J} \bar{u}_t^j \sum_{m=1}^{M} p_t^m l_t^m - (1 - \eta_e) \sum_{t=1}^{T} l_t^{\tilde{m}}$$

$$\leq \frac{2^b(1-\eta_e)}{\epsilon} \ln M + \sum_{t=1}^{T} \sum_{j=1}^{J} \bar{u}_t^j \sum_{m=1}^{M} \frac{(p_t^m - \frac{\eta_e}{M})\epsilon 2^b}{2} \left( \frac{l_t^m}{2^b} \right)^2 + \sum_{t=1}^{T} \sum_{j=1}^{J} \bar{u}_t^j \sum_{m=1}^{M} \frac{\eta_e}{M} l_t^m$$

$$\overset{(i)}{\leq} \frac{2^b}{\epsilon} \ln M + \sum_{t=1}^{T} \sum_{j=1}^{J} \bar{u}_t^j \sum_{m=1}^{M} \frac{p_t^m \epsilon 2^b}{2} \left( \frac{l_t^m}{2^b} \right)^2 + \sum_{t=1}^{T} \sum_{j=1}^{J} \bar{u}_t^j \sum_{m=1}^{M} \frac{\eta_e}{M} l_t^m \tag{41}$$

*where (i) follows $p_t^m \geq \frac{\eta_e}{M}$ since $0 < \eta_e \leq 1$. Recall the probability of observing the loss of m-th model at time t is $q_t^m$. The expected first and second moments of $l_t^m$ given the losses incurred up to time instant $t-1$,*

*i.e,* $\{L\left(\bar{\alpha}_\tau^m, \alpha_\tau^m\right)\}_{\tau=1}^{t-1}$ *can be written as*

$$\mathbb{E}\left[l_t^m\right] = \sum_{j=1}^{J} {p'}_t^j \left(1 - \left(1 - p_t^{mj}\right)^N\right) \frac{L\left(\bar{\alpha}_t^m, \alpha_t^m\right)}{q_t^m} = L\left(\bar{\alpha}_t^m, \alpha_t^m\right)$$

$$\mathbb{E}\left[(l_t^m)^2\right] = \sum_{j=1}^{J} {p'}_t^j \left(1 - \left(1 - p_t^{mj}\right)^N\right) \frac{L^2\left(\bar{\alpha}_t^m, \alpha_t^m\right)}{(q_t^m)^2} = \frac{L^2\left(\bar{\alpha}_t^m, \alpha_t^m\right)}{q_t^m} \le \frac{(1+\eta)^2}{q_t^m} \tag{42}$$

*Taking the expected value of equation 41 at each time t we have*

$$\sum_{t=1}^{T}\sum_{j=1}^{J} \bar{u}_t^j \sum_{m=1}^{M} p_t^m L\left(\bar{\alpha}_t^m, \alpha_t^m\right) \; - \sum_{t=1}^{T} L\left(\bar{\alpha}_t^{\tilde{m}}, \alpha_t^{\tilde{m}}\right)$$

$$\le \frac{2^b}{\epsilon} \ln M + \sum_{t=1}^{T}\sum_{j=1}^{J} \bar{u}_t^j \sum_{m=1}^{M} \frac{\eta_e}{M}(1+\eta) \sum_{t=1}^{T}\sum_{j=1}^{J} \bar{u}_t^j \sum_{m=1}^{M} p_t^m \frac{\epsilon 2^b \left(\frac{(1+\eta)^2}{2^{2b}q_t^m}\right)}{2}$$

$$= \frac{2^b}{\epsilon} \ln M + T\eta_e(\eta+1) + \sum_{t=1}^{T}\sum_{m=1}^{M} p_t^m \epsilon 2^{b-1} \frac{(1+\eta)^2}{2^{2b}q_t^m} \tag{43}$$

*Based on definition of* $q_t^m$, *we obtain*

$$q_t^m = \sum_{j=1}^{J} p_t^j \left(1 - (1 - p_t^m)^N\right) = \sum_{j=1}^{J} p_t^j p_t^m \left(1 + ... + (1 - p_t^m)^{N-1}\right)$$

$$\ge \sum_{j=1}^{J} p_t^j p_t^m = \sum_{j=1}^{J} \bar{u}_t^j p_t^m = p_t^m \tag{44}$$

*So according to equation 44 we have*

$$\sum_{t=1}^{T}\sum_{j=1}^{J} \bar{u}_t^j \sum_{m=1}^{M} p_t^m L\left(\bar{\alpha}_t^m, \alpha_t^m\right) \; - \sum_{t=1}^{T} L\left(\bar{\alpha}_t^{\tilde{m}}, \alpha_t^{\tilde{m}}\right)$$

$$\le \frac{2^b}{\epsilon} \ln M + T\eta_e(\eta+1) + T\epsilon 2^{b-1} M \frac{(1+\eta)^2}{2^{2b}} \tag{45}$$

*According to the procedure of generating the graph* $G_t$ *which is presented in Algorithm 1, for each selective node a subset of models is chosen in N independent trials. In fact, a subset of models is assigned to each node in N independent trials and in each trial one model is assigned and its associated entry in the sub-adjacency matrix becomes 1. Now let* $b_m$ *represents the frequency that m-th model is chosen in N independent trials. Thus,* $\{b_m\}_{m=1}^{M}$ *can be viewed as the solution to the following linear equation*

$$b_1 + b_2 + ... + b_M = N, \;\; s.t. \;\; b_m \ge 0, b_m \in \mathbb{N} \tag{46}$$

*There are* $\binom{M+N-1}{M}$ *different solutions for above equation. Let* $\{b_{m,q}\}_{m=1}^{M}$ *denote the q-th set of solutions. For the expected value of loss we have*

$$\mathbb{E}\left[L\left(\bar{\alpha}_t^m, \alpha_t^m\right)\right] = \sum_{j=1}^{J} \bar{u}_t^j \sum_{q=1}^{\binom{M+N-1}{M}} N! \left(\prod_{m=1}^{M} \frac{(p_t^m)^{b_{m,q}}}{b_{m,q}!}\right) \sum_{m \in S_{t,q}} \bar{w}_t^m L\left(\bar{\alpha}_t^m, \alpha_t^m\right)$$

$$\le \sum_{j=1}^{J} \bar{u}_t^j \sum_{q=1}^{\binom{M+N-1}{M}} N! \left(\prod_{m=1}^{M} \frac{(p_t^m)^{b_{m,q}}}{b_{m,q}!}\right) \sum_{m \in S_{t,q}} L\left(\bar{\alpha}_t^m, \alpha_t^m\right), \tag{47}$$

*Note that the number of ways to solve equation 46 when m-th kernel is chosen at first trial is equals to the number of ways to solve the following problem.*

$$\tilde{b}_{1,m} + \tilde{b}_{2,m} + ... + \tilde{b}_{N,m} = N - 1, \;\; s.t. \;\; \tilde{b}_{n,m} \ge 0, \tilde{b}_{n,m} \in \mathbb{N}. \tag{48}$$

*There are $\binom{M+N-2}{M}$ different solutions for equation 48, Let $\{\tilde{b}^q_{n,m}\}^M_{m=1}$ denotes the q-th set of solution for equation 48. Therefore we can conclude the following quality.*

$$\sum_{j=1}^{J} \bar{u}_t^j \sum_{q=1}^{\binom{M+N-1}{M}} N! \left(\prod_{m=1}^{M} \frac{(p_t^m)^{b_{m,q}}}{b_{m,q}!}\right) \sum_{m \in S_{t,q}} L\left(\bar{\alpha}_t^m, \alpha_t^m\right)$$

$$= \sum_{j=1}^{J} \bar{u}_t^j \sum_{m=1}^{M} p_t^m \sum_{q=1}^{\binom{M+N-2}{M}} N! \left(\prod_{m=1}^{M} \frac{(p_t^m)^{\tilde{b}^q_{n,m}}}{\tilde{b}^q_{n,m}!}\right) L\left(\bar{\alpha}_t^m, \alpha_t^m\right) \tag{49}$$

*where $\sum_{q=1}^{\binom{M+N-2}{M}} N! \left(\prod_{m=1}^{M} \frac{(p_t^m)^{\tilde{b}^q_{n,m}}}{\tilde{b}^q_{n,m}!}\right)$ is the total probability of all $\binom{M+N-2}{M}$ of equation 48. Therefore, $\sum_{q=1}^{\binom{M+N-2}{M}} N! \left(\prod_{m=1}^{M} \frac{(p_t^m)^{\tilde{b}^q_{n,m}}}{\tilde{b}^q_{n,m}!}\right) = 1$. We obtain*

$$\mathbb{E}\left[L\left(\bar{\alpha}_t^m, \alpha_t^m\right)\right] \leq \sum_{j=1}^{J} \bar{u}_t^j \sum_{m=1}^{M} p_t^m L\left(\bar{\alpha}_t^m, \alpha_t^m\right) \tag{50}$$

*So we have*

$$\sum_{t=1}^{T} \mathbb{E}\left[L\left(\bar{\alpha}_t^m, \alpha_t^m\right)\right] - \sum_{t=1}^{T} L\left(\bar{\alpha}_t^{\tilde{m}}, \alpha_t^{\tilde{m}}\right) \leq \frac{2^b}{\epsilon}\ln M + T\eta_e(\eta+1) + T\epsilon 2^{b-1} M \frac{(1+\eta)^2}{2^{2b}} \tag{51}$$

*which concludes to proof of Lemma 2.*

Now, we define the best model in the static environment as

$$m^* = \arg\min_{m \in M} \sum_{t=1}^{T} L(\bar{\alpha}_t^m, \alpha^m).$$

Then, we replace $\tilde{m}$ with best model $m^*$ in Lemma 1 and Lemma 2. Summing results of two lemmas lead to:

$$\sum_{t=1}^{T} \mathbb{E}\left[L\left(\bar{\alpha}_t^m, \alpha_t^m\right)\right] - \sum_{t=1}^{T} L(\bar{\alpha}_t^{m^*}, \alpha^{m^*}) \leq \sqrt{T}\left(\frac{M}{2\eta\eta_e}(1+2\eta)^2 + \frac{\eta}{\alpha}\right)$$

$$+ \frac{2^b}{\epsilon}\ln M + T\eta_e(\eta+1) + T\epsilon 2^{b-1} M \frac{(1+\eta)^2}{2^{2b}}$$

$$\overset{(i)}{\leq} \sqrt{T}\left(\frac{MT^{\frac{1}{4}}}{2\eta}(1+2\eta)^2 + \frac{\eta}{\alpha} + 2^b\ln M + T^{\frac{1}{4}}(\eta+1) + M2^{-b-1}(1+\eta)^2\right) \tag{52}$$

where in $(i)$, we set $\epsilon = \frac{1}{\sqrt{T}}$ and $\eta_e = T^{-\frac{1}{4}}$

**Lemma 3** *By defining $A_t^m := \sum_{q=1}^{Q} N! \left(\prod_{m=1}^{M} \frac{(p_t^m)^{b_{m,q}}}{b_{m,q}!}\right) \bar{w}_t^{mq}$ for any $t \in G_k$, we can say $|A_t^m - A_{t+1}^m| \leq 2(N+1)\epsilon(1+\eta)\frac{M}{\eta_e 2^b}$*

**Proof:** *Based on definition of $A_t^m$ and $A_{t+1}^m$ we have:*

$$|A_t^m - A_{t+1}^m| = \left|\sum_{q=1}^{Q} \frac{N!}{\left(\prod_{m=1}^{M} b_{m,q}!\right)} \left(\left(\prod_{m=1}^{M} (p_t^m)^{b_{m,q}}\right) \bar{w}_t^{mq} - \left(\prod_{m=1}^{M} (p_{t+1}^m)^{b_{m,q}}\right) \bar{w}_{t+1}^{mq}\right)\right|$$

$$\leq \sum_{q=1}^{Q} \frac{N!}{\left(\prod_{m=1}^{M} b_{m,q}!\right)} \left|\left(\prod_{m=1}^{M} (p_t^m)^{b_{m,q}}\right) \bar{w}_t^{mq} - \left(\prod_{m=1}^{M} (p_{t+1}^m)^{b_{m,q}}\right) \bar{w}_{t+1}^{mq}\right|$$

$$\tag{53}$$

*By taking the ln from each term in absolute value of equation 53, the equation can be rewritten as:*

$$
\left| \ln \left( \prod_{m=1}^{M} \left( p_t^m \right)^{b_{m,q}} \right) \bar{w}_t^{mq} - \ln \left( \prod_{m=1}^{M} \left( p_{t+1}^m \right)^{b_{m,q}} \right) \bar{w}_{t+1}^{mq} \right|
$$

$$
= \left| \sum_{m=1}^{M} b_{m,q} \ln p_t^m + \ln \bar{w}_t^{mq} - \sum_{m=1}^{M} b_{m,q} \ln p_{t+1}^m + \ln \bar{w}_{t+1}^{mq} \right|
$$

$$
\leq \sum_{m=1}^{M} b_{m,q} \left| \ln p_t^m - \ln p_{t+1}^m \right| + \left| \ln \bar{w}_t^{mq} - \ln \bar{w}_{t+1}^{mq} \right|
$$

$$
= \sum_{m=1}^{M} b_{m,q} \left| \ln \frac{p_t^m}{p_{t+1}^m} \right| + \left| \ln \frac{\bar{w}_t^{mq}}{\bar{w}_{t+1}^{mq}} \right| = \sum_{m=1}^{M} b_{m,q} \left| \ln \frac{(1-\eta_e)\frac{w_t^m}{W_t} + \frac{\eta_e}{M}}{(1-\eta_e)\frac{w_{t+1}^m}{W_{t+1}} + \frac{\eta_e}{M}} \right| + \left| \ln \frac{w_t^{mq} W_{t+1}^q}{w_{t+1}^{mq} W_t^q} \right|
$$

$$
\tag{54}
$$

*According to the update rule equation 7, we have*

$$
= \sum_{m=1}^{M} b_{m,q} \left| \ln \frac{\frac{M(1-\eta_e)w_t^m + \eta_e W_t}{W_t M}}{\frac{M(1-\eta_e)w_{t+1}^m + \eta_e W_{t+1}}{W_{t+1} M}} \right| + \left| \ln \frac{w_t^{mq} W_{t+1}^q}{w_t^{mq} \exp\left(-\epsilon \frac{l_t^m}{2^b}\right) W_t^q} \right|
$$

$$
= \sum_{m=1}^{M} b_{m,q} \left| \ln \frac{W_{t+1}}{W_t} + \ln \frac{M(1-\eta_e)w_t^m + \eta_e W_t}{M(1-\eta_e)w_{t+1}^m + \eta_e W_{t+1}} \right| + \left| \ln \frac{W_{t+1}^q}{W_t^q} - \epsilon \frac{l_t^m}{2^b} \right|
$$

$$
\leq \sum_{m=1}^{M} b_{m,q} \left| \ln \frac{W_{t+1}}{W_t} \right| + \sum_{m=1}^{M} b_{m,q} \left| \ln \frac{M(1-\eta_e)w_t^m + \eta_e W_t}{M(1-\eta_e)w_{t+1}^m + \eta_e W_{t+1}} \right| + \left| \ln \frac{W_{t+1}^q}{W_t^q} \right| + \epsilon \frac{l_t^m}{2^b}
$$

$$
\tag{55}
$$

$$
= \sum_{m=1}^{M} b_{m,q} \left| \ln \frac{\sum_{\tilde{m}=1}^{M} w_t^{\tilde{m}} \exp\left(-\epsilon \frac{l_t^{\tilde{m}}}{2^b}\right)}{\sum_{\tilde{m}=1}^{M} w_t^{\tilde{m}}} \right| + \left| \ln \frac{\sum_{\tilde{m} \in q} w_t^{\tilde{m}} \exp\left(-\epsilon \frac{l_t^{\tilde{m}}}{2^b}\right)}{\sum_{\tilde{m} \in q} w_t^{\tilde{m}}} \right|
$$

$$
+ \sum_{m=1}^{M} b_{m,q} \left| \ln \frac{M(1-\eta_e)w_t^m + \eta_e \sum_{\tilde{m}=1}^{M} w_t^{\tilde{m}}}{M(1-\eta_e)w_t^m \exp\left(-\epsilon \frac{l_t^m}{2^b}\right) + \eta_e \sum_{\tilde{m}=1}^{M} w_t^{\tilde{m}} \exp\left(-\epsilon \frac{l_t^{\tilde{m}}}{2^b}\right)} \right| + \epsilon \frac{l_t^m}{2^b}
$$

$$
\overset{(i)}{\leq} \sum_{m=1}^{M} b_{m,q} \left| \ln \frac{\sum_{\tilde{m}=1}^{M} w_t^{\tilde{m}} \exp\left(-f\left(\epsilon\right)\right)}{\sum_{\tilde{m}=1}^{M} w_t^{\tilde{m}}} \right| + \left| \ln \frac{\sum_{\tilde{m} \in q} w_t^{\tilde{m}} \exp\left(-f\left(\epsilon\right)\right)}{\sum_{\tilde{m} \in q} w_t^{\tilde{m}}} \right|
$$

$$
+ \sum_{m=1}^{M} b_{m,q} \left| \ln \frac{M(1-\eta_e)w_t^m + \eta_e \sum_{\tilde{m}=1}^{M} w_t^{\tilde{m}}}{M(1-\eta_e)w_t^m \exp\left(-f\left(\epsilon\right)\right) + \eta_e \sum_{\tilde{m}=1}^{M} w_t^{\tilde{m}} \exp\left(-f\left(\epsilon\right)\right)} \right| + f\left(\epsilon\right)
$$

$$
= 2 \sum_{m=1}^{M} b_{m,q} \left| f(\epsilon) \right| + \left| f(\epsilon) \right| + f(\epsilon) = 2(N+1)f(\epsilon) \overset{(ii)}{=} 2(N+1)\epsilon(1+\eta) \frac{M}{\eta_e 2^b} \tag{56}
$$

Note that in $(i)$ we replace the $\epsilon^{\frac{l_t^m}{2^b}}$ terms with their maximum value $f(\epsilon)$. $(ii)$ follows equation 33 and the fact that maximum length of every prediction set would be $K$. By defining $B_1 := 2\frac{M}{\eta_e 2^b}(1+\eta)(N+1)$ we have:

$$
\left| \ln \left( \prod_{m=1}^{M} (p_t^m)^{b_{m,q}} \right) \bar{w}_t^{mq} - \ln \left( \prod_{m=1}^{M} (p_{t+1}^m)^{b_{m,q}} \right) \bar{w}_{t+1}^{mq} \right| <= B_1 \epsilon
$$

$$
\sum_{q=1}^{Q} \frac{N!}{\left( \prod_{m=1}^{M} b_{m,q}! \right)} \left| \left( \prod_{m=1}^{M} (p_t^m)^{b_{m,q}} \right) \bar{w}_t^{mq} - \left( \prod_{m=1}^{M} (p_{t+1}^m)^{b_{m,q}} \right) \bar{w}_{t+1}^{mq} \right| \overset{(ii)}{<=}
$$

$$
\sum_{q=1}^{Q} \frac{N!}{\left( \prod_{m=1}^{M} b_{m,q}! \right)} \left( e^{B_1 \epsilon} - 1 \right) \overset{(iii)}{=} B_2 \left( e^{C\epsilon} - 1 \right), \tag{57}
$$

where $(ii)$ follow $|x - y| <= e^C - 1$ if $|\ln x - \ln y| <= C, x, y[0,1]$. $(iii)$ follows the definition $B_2 := \sum_{q=1}^{Q} \frac{N!}{\left( \prod_{m=1}^{M} b_{m,q}! \right)}$.

### A.3 Proof of Theorem 3

The proof of coverage for EGMOCP follows the same steps as the proof for the GMOCP algorithm up to the point where the upper bound in equation 25 is derived using Lemma 3. At this stage, we instead apply Lemma 5, where the constant $C_1$ replace the original constant $B_1$. As a result, the coverage error for EGMOCP is bounded as follows:

$$
\left| \frac{1}{T} \sum_{t=1}^{T} \sum_{q=1}^{Q=\binom{M+N-1}{M}} N! \left( \prod_{m=1}^{M} \frac{(p_t^m)^{b_{m,q}}}{b_{m,q}!} \right) \sum_{m \in S_{t,q}} \bar{w}_t^{mq} a_t^m \right|
$$

$$
\leq T^{-\frac{1}{4}} \left( 2M + \frac{2\sqrt{2}M(1+\eta)}{\eta} + \frac{2M(1+\eta)}{\eta} B_2 C_1 (1+o(1)) + \frac{M}{\alpha^3} \log T \right). \tag{58}
$$

### A.4 Proof of Theorem 4

To prove regret bound for EGMOCP, we first establish the following lemma.

**Lemma 4** *for any model $\tilde{m} \in [M]$ following bound holds*

$$
\sum_{t=1}^{T} \mathbb{E}\left[ L\left( \bar{\alpha}_\tau^m, \alpha_\tau^m \right) \right] - \sum_{t=1}^{T} L\left( \bar{\alpha}_\tau^{\tilde{m}}, \alpha_\tau^{\tilde{m}} \right)
$$

$$
\leq \frac{2^b}{\epsilon(1-\beta)} \ln M + \frac{\beta 2^b}{(1-\beta)} N_{labels} T + MT \frac{\epsilon 2^{b-1}}{(1-\beta)} \left( (1-\beta)^2 \frac{(1+\eta)^2}{2^{2b}} + (1-\beta)\beta \frac{(1+\eta)N_{labels}}{2^{b-1}} + (N_{labels}\beta)^2 \right) \tag{59}
$$

**Proof:** *Defining $W_t := \sum_{m=1}^{M} w_t^m$ and $\bar{u}_t^j := \frac{u_t^j}{U_t}$, we have*

$$
\frac{W_{t+1}}{W_t} = \sum_{j=1}^{J} \bar{u}_t^j \frac{W_{t+1}}{W_t} = \sum_{j=1}^{J} \bar{u}_t^j \sum_{m=1}^{M} \frac{w_{t+1}^m}{W_t}
$$

$$
= \sum_{j=1}^{J} \bar{u}_t^j \sum_{m=1}^{M} \frac{w_t^m}{W_t} \exp\left( -\epsilon \left[ (1-\beta)\frac{l_t^m}{2^b} + \beta Len(\alpha_t^m)\mathbb{I}\{m \in S_t\} \right] \right)
$$

$$
= \sum_{j=1}^{J} \bar{u}_t^j \sum_{m=1}^{M} \frac{p_t^m - \frac{\eta_e}{M}}{1 - \eta_e} \exp\left( -\epsilon \left[ (1-\beta)\frac{l_t^m}{2^b} + \beta Len(\alpha_t^m)\mathbb{I}\{m \in S_t\} \right] \right) \tag{60}
$$

*Using the inequality* $\exp(-x) \leq 1 - x + \frac{x^2}{2}, \forall x \geq 0$ *leads to*

$$\frac{W_{t+1}}{W_t} \leq \sum_{j=1}^{J} \bar{u}_t^j \sum_{m=1}^{M} \frac{p_t^m - \frac{\eta_e}{M}}{1 - \eta_e} \left(1 - \epsilon\left((1-\beta)\frac{l_t^m}{2^b} + \beta Len(\alpha_t^m)\mathbb{I}\{m \in S_t\}\right)\right)$$
$$+ \sum_{j=1}^{J} \bar{u}_t^j \sum_{m=1}^{M} \frac{p_t^m - \frac{\eta_e}{M}}{1 - \eta_e} \left(\frac{\epsilon^2\left((1-\beta)\frac{l_t^m}{2^b} + \beta Len(\alpha_t^m)\mathbb{I}\{m \in S_t\}\right)^2}{2}\right) \tag{61}$$

*By taking the logarithm from both sides of above inequality we have*

$$\ln \frac{W_{t+1}}{W_t} \leq \ln \sum_{j=1}^{J} \bar{u}_t^j \sum_{m=1}^{M} \frac{p_t^m - \frac{\eta_e}{M}}{1 - \eta_e} (1 - \epsilon((1-\beta)\frac{l_t^m}{2^b} + \beta Len(\alpha_t^m)\mathbb{I}\{m \in S_t\})$$
$$+ \frac{\epsilon^2\left((1-\beta\frac{l_t^m}{2^b} + \beta Len(\alpha_t^m)\mathbb{I}\{m \in S_t\}\right)^2}{2})$$
$$\overset{(i)}{\leq} \sum_{j=1}^{J} \bar{u}_t^j \sum_{m=1}^{M} \frac{p_t^m - \frac{\eta_e}{M}}{1 - \eta_e} \left(1 - \epsilon\left((1-\beta)\frac{l_t^m}{2^b} + \beta Len(\alpha_t^m)\mathbb{I}\{m \in S_t\}\right)\right)$$
$$+ \sum_{j=1}^{J} \bar{u}_t^j \sum_{m=1}^{M} \frac{p_t^m - \frac{\eta_e}{M}}{1 - \eta_e} \left(\frac{\epsilon^2\left((1-\beta)\frac{l_t^m}{2^b} + \beta Len(\alpha_t^m)\mathbb{I}\{m \in S_t\}\right)^2}{2}\right) - 1$$
$$\overset{(ii)}{=} \sum_{j=1}^{J} \bar{u}_t^j \sum_{m=1}^{M} \frac{p_t^m - \frac{\eta_e}{M}}{1 - \eta_e} \left(-\epsilon\left((1-\beta)\frac{l_t^m}{2^b} + \beta Len(\alpha_t^m)\mathbb{I}\{m \in S_t\}\right)\right)$$
$$+ \sum_{j=1}^{J} \bar{u}_t^j \sum_{m=1}^{M} \frac{p_t^m - \frac{\eta_e}{M}}{1 - \eta_e} \left(\frac{\epsilon^2\left((1-\beta)\frac{l_t^m}{2^b} + \beta Len(\alpha_t^m)\mathbb{I}\{m \in S_t\}\right)^2}{2}\right) \tag{62}$$

*where* $\overset{(i)}{\leq}$ *follows* $1 + x \leq \exp x$ *in case we replace* $x$ *with* $\ln y$ *which leads to* $1 + \ln y \leq y$, *and* $\overset{(ii)}{\leq}$ *follows* $\sum_{j=1}^{J} \bar{u}_t^j \sum_{m=1}^{M} \frac{p_t^m - \frac{\eta_e}{M}}{1 - \eta_e} = 1$. *Summing equation 62 over* $t$ *from 1 to* $T$ *result in*

$$\ln \frac{W_{T+1}}{W_1} \leq \sum_{t=1}^{T}\sum_{j=1}^{J} \bar{u}_t^j \sum_{m=1}^{M} \frac{p_t^m - \frac{\eta_e}{M}}{1 - \eta_e} \left(-\epsilon\left((1-\beta)\frac{l_t^m}{2^b} + \beta Len(\alpha_t^m)\mathbb{I}\{m \in S_t\}\right)\right)$$
$$+ \sum_{t=1}^{T}\sum_{j=1}^{J} \bar{u}_t^j \sum_{m=1}^{M} \frac{p_t^m - \frac{\eta_e}{M}}{1 - \eta_e} \left(\frac{\epsilon^2\left((1-\beta)\frac{l_t^m}{2^b} + \beta Len(\alpha_t^m)\mathbb{I}\{m \in S_t\}\right)^2}{2}\right) \tag{63}$$

*Furthermore, recall the updating rule of* $w_t^m$ *in equation 17, for any model* $\tilde{m} \in [M]$ *we have*

$$\ln \frac{W_{T+1}}{W_1} \geq \ln \frac{w_{T+1}^{\tilde{m}}}{W_1} = \ln w_1^{\tilde{m}} \exp\left(\sum_{t=1}^{T} -\epsilon\left((1-\beta)\frac{l_t^{\tilde{m}}}{2^b} + \beta Len(\alpha_t^{\tilde{m}})\mathbb{I}\{\tilde{m} \in S_t\}\right)\right) - \ln 1$$
$$= -\ln M - \sum_{t=1}^{T} \epsilon\left((1-\beta)\frac{l_t^{\tilde{m}}}{2^b} + \beta Len(\alpha_t^{\tilde{m}})\mathbb{I}\{\tilde{m} \in S_t\}\right) \tag{64}$$

*combining equation 63 with equation 64 result in*

$$\sum_{t=1}^{T}\sum_{j=1}^{J}\bar{u}_t^j\sum_{m=1}^{M}\frac{p_t^m-\frac{\eta_e}{M}}{1-\eta_e}\left(-\epsilon\left((1-\beta)\frac{l_t^m}{2^b}+\beta Len(\alpha_t^m)\mathbb{I}\{m\in S_t\}\right)\right)$$

$$+\sum_{t=1}^{T}\sum_{j=1}^{J}\bar{u}_t^j\sum_{m=1}^{M}\frac{p_t^m-\frac{\eta_e}{M}}{1-\eta_e}\left(\frac{\epsilon^2\left((1-\beta)\frac{l_t^m}{2^b}+\beta Len(\alpha_t^m)\mathbb{I}\{m\in S_t\}\right)^2}{2}\right)$$

$$\geq-\ln M-\sum_{t=1}^{T}\epsilon\left((1-\beta)\frac{l_t^{\tilde{m}}}{2^b}+\beta Len(\alpha_t^{\tilde{m}})\mathbb{I}\{\tilde{m}\in S_t\}\right) \tag{65}$$

*Multiplying $\frac{2^b(1-\eta_e)}{\epsilon(1-\beta)}$ to both sides and rearrangement leads to*

$$\sum_{t=1}^{T}\sum_{j=1}^{J}\bar{u}_t^j\sum_{m=1}^{M}p_t^m l_t^m-(1-\eta_e)\sum_{t=1}^{T}l_t^{\tilde{m}}\leq\frac{2^b(1-\eta_e)}{\epsilon(1-\beta)}\ln M$$

$$+\sum_{t=1}^{T}\frac{\beta 2^b(1-\eta_e)}{1-\beta}Len(\alpha_t^{\tilde{m}})\mathbb{I}\{\tilde{m}\in S_t\}-\sum_{t=1}^{T}\sum_{j=1}^{J}\bar{u}_t^j\sum_{m=1}^{M}\frac{2^b\beta(p_t^m-\frac{\eta_e}{M})}{1-\beta}Len(\alpha_t^m)\mathbb{I}\{m\in S_t\}$$

$$+\sum_{t=1}^{T}\sum_{j=1}^{J}\bar{u}_t^j\sum_{m=1}^{M}\frac{(p_t^m-\frac{\eta_e}{M})\epsilon 2^b}{2(1-\beta)}\left((1-\beta)\frac{l_t^m}{2^b}+\beta Len(\alpha_t^m)\mathbb{I}\{m\in S_t\}\right)^2+\sum_{t=1}^{T}\sum_{j=1}^{J}\bar{u}_t^j\sum_{m=1}^{M}\frac{\eta_e}{M}l_t^m$$

$$\overset{(i)}{\leq}\frac{2^b}{\epsilon(1-\beta)}\ln M+\sum_{t=1}^{T}\frac{\beta 2^b}{1-\beta}Len(\alpha_t^{\tilde{m}})\mathbb{I}\{\tilde{m}\in S_t\}$$

$$+\sum_{t=1}^{T}\sum_{j=1}^{J}\bar{u}_t^j\sum_{m=1}^{M}\frac{p_t^m\epsilon 2^b}{2(1-\beta)}\left((1-\beta)\frac{l_t^m}{2^b}+\beta Len(\alpha_t^m)\mathbb{I}\{m\in S_t\}\right)^2+\sum_{t=1}^{T}\sum_{j=1}^{J}\bar{u}_t^j\sum_{m=1}^{M}\frac{\eta_e}{M}l_t^m \tag{66}$$

*Taking the expected value of equation 66 at each time t we have*

$$\sum_{t=1}^{T}\sum_{j=1}^{J}\bar{u}_t^j\sum_{m=1}^{M}p_t^m L\left(\bar{\alpha}_t^m,\alpha_t^m\right)-\sum_{t=1}^{T}L\left(\bar{\alpha}_t^{\tilde{m}},\alpha_t^{\tilde{m}}\right)$$

$$\leq\frac{2^b}{\epsilon(1-\beta)}\ln M+\sum_{t=1}^{T}\frac{\beta 2^b}{1-\beta}Len(\alpha_t^{\tilde{m}})q_t^{\tilde{m}}+\sum_{t=1}^{T}\sum_{j=1}^{J}\bar{u}_t^j\sum_{m=1}^{M}\frac{\eta_e}{M}(1+\eta)$$

$$+\sum_{t=1}^{T}\sum_{j=1}^{J}\bar{u}_t^j\sum_{m=1}^{M}p_t^m\frac{\epsilon 2^b\left((1-\beta)^2\frac{(1+\eta)^2}{2^{2b}q_t^m}+2(1-\beta)\beta\frac{(1+\eta)Len(\alpha_t^m)q_t^m}{2^b}+(\beta)^2 Len^2(\alpha_t^m)q_t^m\right)}{2(1-\beta)}$$

$$\leq\frac{2^b}{\epsilon(1-\beta)}\ln M+\frac{\beta 2^b}{1-\beta}N_{labels}T+T\eta_e(\eta+1)$$

$$+\sum_{t=1}^{T}\sum_{m=1}^{M}p_t^m\frac{\epsilon 2^{b-1}}{1-\beta}\left((1-\beta)^2\frac{(1+\eta)^2}{2^{2b}q_t^m}+(1-\beta)\beta\frac{(1+\eta)N_{labels}}{2^{b-1}}+(N_{labels}\beta)^2\right) \tag{67}$$

*So according to equation 44 we have*

$$\sum_{t=1}^{T}\sum_{j=1}^{J}\bar{u}_t^j\sum_{m=1}^{M}p_t^m L\left(\bar{\alpha}_t^m,\alpha_t^m\right)-\sum_{t=1}^{T}L\left(\bar{\alpha}_t^{\tilde{m}},\alpha_t^{\tilde{m}}\right)$$

$$\leq\frac{2^b}{\epsilon(1-\beta)}\ln M+\frac{\beta 2^b}{1-\beta}N_{labels}T+T\eta_e(\eta+1)+T\frac{\epsilon 2^{b-1}}{1-\beta}\left((1-\beta)^2 M\frac{(1+\eta)^2}{2^{2b}}+(1-\beta)\beta\frac{(1+\eta)N_{labels}}{2^{b-1}}+(N_{labels}\beta)^2\right) \tag{68}$$

*By following the same steps as equation 46-equation 49, we obtain*

$$\mathbb{E}\left[L\left(\bar{\alpha}_t^m, \alpha_t^m\right)\right] \leq \sum_{j=1}^{J} \bar{u}_t^j \sum_{m=1}^{M} p_t^m L\left(\bar{\alpha}_t^m, \alpha_t^m\right) \tag{69}$$

*So we have*

$$
\begin{aligned}
\sum_{t=1}^{T} \mathbb{E}\left[L\left(\bar{\alpha}_t^m, \alpha_t^m\right)\right] - \sum_{t=1}^{T} L\left(\bar{\alpha}_t^{\tilde{m}}, \alpha_t^{\tilde{m}}\right) &\leq \frac{2^b}{\epsilon(1-\beta)} \ln M \\
+ \frac{\beta 2^b}{1-\beta} N_{labels} T + T\eta_e(\eta+1) + T\frac{\epsilon 2^{b-1}}{1-\beta} &\left((1-\beta)^2 M \frac{(1+\eta)^2}{2^{2b}} + (1-\beta)\beta\frac{(1+\eta)N_{labels}}{2^{b-1}} + (N_{labels}\beta)^2\right)
\end{aligned}
\tag{70}
$$

*which concludes to proof of Lemma equation 4.*

Then, we replace $\tilde{m}$ with best model $m^*$ in Lemma 1 and Lemma 4. Summing results of two lemmas lead to:

$$
\begin{aligned}
\sum_{t=1}^{T} \mathbb{E}\left[L\left(\bar{\alpha}_t^m, \alpha_t^m\right)\right] - \sum_{t=1}^{T} L(\bar{\alpha}_t^{m^*}, \alpha^{m^*}) &\leq \sqrt{T}\left(\frac{M}{2\eta\eta_e}(1+2\eta)^2 + \frac{\eta}{\alpha}\right) \\
+ \frac{2^b}{\epsilon(1-\beta)}\ln M + \frac{\beta 2^b}{1-\beta}N_{\text{labels}}T + T\eta_e(\eta+1)+ & \\
T\frac{\epsilon 2^{b-1}}{1-\beta}\left((1-\beta)^2 M\frac{(1+\eta)^2}{2^{2b}} + (1-\beta)\beta\frac{(1+\eta)N_{\text{labels}}}{2^{b-1}} + (N_{\text{labels}}\beta)^2\right) & \\
\overset{(i)}{\leq} \sqrt{T}\left(\frac{MT^{\frac{1}{4}}}{2\eta}(1+2\eta)^2 + \frac{\eta}{\alpha}\right) & \\
+ \sqrt{T}\left(\frac{\sqrt{T}}{\sqrt{T}-1}2^b\ln M + \frac{\sqrt{T}}{\sqrt{T}-1}2^b N_{\text{labels}} + T^{\frac{1}{4}}(\eta+1) + \frac{\sqrt{T}-1}{\sqrt{T}}\frac{M(1+\eta)^2}{2^{b+1}} + \frac{(1+\eta)N_{\text{labels}}}{\sqrt{T}} + \frac{1}{T-\sqrt{T}}2^{b-1}N_{\text{labels}}^2\right) &
\end{aligned}
\tag{71}
$$

where in $(i)$, we set $\epsilon = \beta = \frac{1}{\sqrt{T}}$ and $\eta_e = T^{-\frac{1}{4}}$

**Lemma 5** *By defining $A_t^m := \sum_{q=1}^{Q} N!\left(\prod_{m=1}^{M} \frac{(p_t^m)^{b_{m,q}}}{b_{m,q}!}\right)\bar{w}_t^{mq}$ for any $t \in G_k$, we can say $|A_t^m - A_{t+1}^m| \leq 2(N+1)\epsilon\left((1-\beta)(1+\eta)\frac{M}{\eta_e 2^b} + \beta K\right)$*

**Proof of Lemma 5** The proof begins with the same steps as in the proof of Lemma 3, up to equation equation 54. However, since the weight update rule in EGMOCP differs from that in GMOCP, we proceed

by applying the update rule defined in equation 17. Therefore, we have:

$$\left| \ln \left( \prod_{m=1}^{M} (p_t^m)^{b_{m,q}} \right) \bar{w}_t^{mq} - \ln \left( \prod_{m=1}^{M} (p_{t+1}^m)^{b_{m,q}} \right) \bar{w}_{t+1}^{mq} \right|$$

$$\leq \sum_{m=1}^{M} b_{m,q} \left| \ln \frac{p_t^m}{p_{t+1}^m} \right| + \left| \ln \frac{\bar{w}_t^{mq}}{\bar{w}_{t+1}^{mq}} \right| = \sum_{m=1}^{M} b_{m,q} \left| \ln \frac{(1-\eta_e)\frac{w_t^m}{W_t} + \frac{\eta_e}{M}}{(1-\eta_e)\frac{w_{t+1}^m}{W_{t+1}} + \frac{\eta_e}{M}} \right| + \left| \ln \frac{w_t^{mq} W_{t+1}^q}{w_{t+1}^{mq} W_t^q} \right|$$

$$= \sum_{m=1}^{M} b_{m,q} \left| \ln \frac{\frac{M(1-\eta_e)w_t^m + \eta_e W_t}{W_t M}}{\frac{M(1-\eta_e)w_{t+1}^m + \eta_e W_{t+1}}{W_{t+1} M}} \right| + \left| \ln \frac{w_t^{mq} W_{t+1}^q}{w_t^{mq} \exp\left(-\epsilon\left((1-\beta)\frac{l_t^m}{2^b} + \beta Len\left(\alpha_t^m\right)\right)\right) W_t^q} \right|$$

$$\leq \sum_{m=1}^{M} b_{m,q} \left| \ln \frac{W_{t+1}}{W_t} \right| + \sum_{m=1}^{M} b_{m,q} \left| \ln \frac{M(1-\eta_e)w_t^m + \eta_e W_t}{M(1-\eta_e)w_{t+1}^m + \eta_e W_{t+1}} \right|$$

$$+ \left| \ln \frac{W_{t+1}^q}{W_t^q} \right| + \epsilon\left((1-\beta)\frac{l_t^m}{2^b} + \beta Len\left(\alpha_t^m\right)\right) \tag{72}$$

By expressing weights at time $t+1$ in terms of its value at time $t$, we have:

$$= \sum_{m=1}^{M} b_{m,q} \left| \ln \frac{\sum_{\tilde{m}=1}^{M} w_t^{\tilde{m}} \exp\left(-\epsilon\left((1-\beta)\frac{l_t^{\tilde{m}}}{2^b} + \beta Len\left(\alpha_t^{\tilde{m}}\right)\right)\right)}{\sum_{\tilde{m}=1}^{M} w_t^{\tilde{m}}} \right|$$

$$+ \sum_{m=1}^{M} b_{m,q} \left| \ln \frac{M(1-\eta_e)w_t^m + \eta_e \sum_{\tilde{m}=1}^{M} w_t^{\tilde{m}}}{M(1-\eta_e)w_t^m \exp\left(-\epsilon\left((1-\beta)\frac{l_t^m}{2^b} + \beta Len\left(\alpha_t^m\right)\right)\right) + \eta_e \sum_{\tilde{m}=1}^{M} w_t^{\tilde{m}} \exp\left(-\epsilon\left((1-\beta)\frac{l_t^{\tilde{m}}}{2^b} + \beta Len\left(\alpha_t^{\tilde{m}}\right)\right)\right)} \right|$$

$$+ \left| \ln \frac{\sum_{\tilde{m}\in q} w_t^{\tilde{m}} \exp\left(-\epsilon\left((1-\beta)\frac{l_t^{\tilde{m}}}{2^b} + \beta Len\left(\alpha_t^{\tilde{m}}\right)\right)\right)}{\sum_{\tilde{m}\in q} w_t^{\tilde{m}}} \right| + \epsilon\left((1-\beta)\frac{l_t^m}{2^b} + \beta Len\left(\alpha_t^m\right)\right)$$

$$\overset{(i)}{\leq} \sum_{m=1}^{M} b_{m,q} \left| \ln \frac{\sum_{\tilde{m}=1}^{M} w_t^{\tilde{m}} \exp\left(-f\left(\epsilon\right)\right)}{\sum_{\tilde{m}=1}^{M} w_t^{\tilde{m}}} \right| + \left| \ln \frac{\sum_{\tilde{m}\in q} w_t^{\tilde{m}} \exp\left(-f\left(\epsilon\right)\right)}{\sum_{\tilde{m}\in q} w_t^{\tilde{m}}} \right|$$

$$+ \sum_{m=1}^{M} b_{m,q} \left| \ln \frac{M(1-\eta_e)w_t^m + \eta_e \sum_{\tilde{m}=1}^{M} w_t^{\tilde{m}}}{M(1-\eta_e)w_t^m \exp\left(-f\left(\epsilon\right)\right) + \eta_e \sum_{\tilde{m}=1}^{M} w_t^{\tilde{m}} \exp\left(-f\left(\epsilon\right)\right)} \right| + f\left(\epsilon\right)$$

$$= 2 \sum_{m=1}^{M} b_{m,q} |f(\epsilon)| + |f(\epsilon)| + f(\epsilon) = 2(N+1)f(\epsilon) \overset{(ii)}{=} 2(N+1)\epsilon\left((1-\beta)(1+\eta)\frac{M}{\eta_e 2^b} + \beta K\right) \tag{73}$$

Note that in $(i)$ we replace the $\epsilon\left((1-\beta)\frac{l_t^m}{2^b} + \beta Len\left(\alpha_t^m\right)\right)$ terms with their maximum value $f(\epsilon)$. $(ii)$ follows equation 33 and the fact that maximum length of every prediction set would be K. By defining $C_1 := 2(N+1)\left((1-\beta)(1+\eta)\frac{M}{\eta_e 2^b} + \beta K\right)$ we have:

$$\left| \ln \left( \prod_{m=1}^{M} (p_t^m)^{b_{m,q}} \right) \bar{w}_t^{mq} - \ln \left( \prod_{m=1}^{M} (p_{t+1}^m)^{b_{m,q}} \right) \bar{w}_{t+1}^{mq} \right| <= C_1 \epsilon$$

$$\sum_{q=1}^{Q} \frac{N!}{\left(\prod_{m=1}^{M} b_{m,q}!\right)} \left| \left( \prod_{m=1}^{M} (p_t^m)^{b_{m,q}} \right) \bar{w}_t^{mq} - \left( \prod_{m=1}^{M} (p_{t+1}^m)^{b_{m,q}} \right) \bar{w}_{t+1}^{mq} \right| \overset{(i)}{<=}$$

$$\sum_{q=1}^{Q} \frac{N!}{\left(\prod_{m=1}^{M} b_{m,q}!\right)} \left(e^{C_1 \epsilon} - 1\right) \overset{(ii)}{=} B_2 \left(e^{C_1 \epsilon} - 1\right) \tag{74}$$

where $(i)$ follows $|x - y| <= e^C - 1$ if $|\ln x - \ln y| <= C, x, y[0,1]$. $(ii)$ follows the definition $B_2 := \sum_{q=1}^{Q} \frac{N!}{\left(\prod_{m=1}^{M} b_{m,q}!\right)}$.

### A.5 Set size Comparison between GMOCP and COMA

According to lemma 6, by replacing the loss function in equation 3 with $Len(\alpha_m^t)$ we derive an upper bound for the prediction set size. We find that the average size of the prediction set at each time $t$ constructed by GMOCP, is smaller than the weighted average across all candidate models. The COMA algorithm yields a prediction set size that is never larger than twice the weighted average size of candidate models ,according to Lemma 2.4 in (Gasparin & Ramdas, 2024a). Therefore, the maximum prediction set size of GMOCP at each time $t$ is half that of COMA.

**Lemma 6** *Expected size of prediction set constructed by GMOCP at each time $t$ is smaller than or equal to the weighted average of the sizes of the prediction sets corresponding to all candidate models.*

$$\mathbb{E}\left[ Len\left(\alpha_t^{\hat{m}}\right)\right] \leq \sum_{m=1}^{M} p_t^m Len\left(\alpha_t^m\right) \tag{75}$$

**Proof of Lemma 6:** According to equation 50, which demonstrates that the expected loss of the online algorithm is smaller than or equal to the weighted average over all models' prediction sets, we can replace $L\left(\bar{\alpha}_t^m, \alpha_t^m\right)$ with $Len(\alpha_m^t)$. Thus, it can be concluded that the expected prediction set size at each time $t$ is smaller than the weighted average of all models.

## B Additional Experiments

### B.1 Equal Number of Models From Each Training Setting

To evaluate the performance of the proposed methods in scenarios with more weak-performing learning models, we conduct additional experiments using a new set of candidate models that includes three versions each of DenseNet121, GoogLeNet, ResNet-18, and ResNet-50—amounting to a total of 12 models. For these experiments, we use the CIFAR-100C dataset under a gradual distribution shift. The results are reported in Table 3 for $J \in \{1, 2, 4\}$ and $N \in \{1, 3, 5, 7\}$. It can be observed that, across all settings, GMOCP constructs smaller prediction sets in less time while satisfying the desired coverage compared to MOCP. Additionally, EGMOCP achieves the smallest prediction sets compared to all benchmarks.

### B.2 CIFAR-10C Dataset

For cases where different selective nodes have equal exploration ratios, experiments on CIFAR-10C with gradual distribution shifts are provided. The results, presented in Table 4, show that GMOCP achieves the fastest run time, while EGMOCP constructs the smallest prediction sets.

### B.3 TinyImageNet Dataset

Here, we conduct experiments on a new dataset featuring a gradual distribution shift using TinyImageNet-C, a corrupted version of the TinyImageNet dataset (Le & Yang, 2015) that contains 200 distinct classes. For this experiment, the hyperparameters $\xi$ and $k_{reg}$ are set to 0.01 and 20, respectively. Additionally, the number of sequential data points is 2500, $\epsilon = 0.1$, and $\beta = 0.02$. Note that since the number of candidate labels is large (200), we report the proportion of prediction sets that include the true label and have a size smaller than 40, instead of focusing solely on prediction sets of length 1. Table 5 shows that GMOCP consistently achieves smaller prediction sets in shorter time compared to MOCP. Additionally, EGMOCP is able to generate smaller prediction sets than both MOCP and GMOCP across all settings.

### B.4 Synthetic Dataset

Additional experiments is conducted sing synthetic data generated in (Hajihashemi & Shen, 2024), which creates distribution shifts using two distinct transformation sequences. From each sequence, two datasets are generated with random variations to ensure uniqueness across samples. Each dataset contains 3,000

Table 3: Results on the CIFAR-100C dataset under gradual distribution shifts and utilizing 12 learning models, evaluated across different values of $N$ and $J$. The target coverage is 90%. Bold numbers denote the best results in each column. GMOCP consistently achieves faster runtime compared to MOCP across all settings. EGMOCP constructs smaller prediction sets and a higher proportion of single-width sets.

| $N$ | $J$ | Method | Coverage (%) | Avg Width | Run Time | Single Width |
|---|---|---|---|---|---|---|
| | | MOCP | $89.98 \pm 0.26$ | $27.28 \pm 4.34$ | $13.11 \pm 0.15$ | $8.25 \pm 1.93$ |
| | | COMA | $\mathbf{90.01 \pm 0.01}$ | $7.12 \pm 0.54$ | $16.89 \pm 0.11$ | $\mathbf{27.39 \pm 0.69}$ |
| 1 | 1 | GMOCP | $88.92 \pm 0.19$ | $25.41 \pm 3.12$ | $\mathbf{5.80 \pm 0.02}$ | $9.22 \pm 1.15$ |
| | | EGMOCP | $88.98 \pm 0.22$ | $10.86 \pm 0.26$ | $7.02 \pm 0.03$ | $20.32 \pm 0.53$ |
| | 2 | GMOCP | $89.13 \pm 0.32$ | $22.92 \pm 1.05$ | $6.02 \pm 0.04$ | $11.08 \pm 0.80$ |
| | | EGMOCP | $89.12 \pm 0.14$ | $11.78 \pm 0.26$ | $7.29 \pm 0.02$ | $19.05 \pm 0.70$ |
| | 4 | GMOCP | $88.94 \pm 0.17$ | $22.46 \pm 0.58$ | $6.31 \pm 0.04$ | $11.54 \pm 0.43$ |
| | | EGMOCP | $88.97 \pm 0.25$ | $11.20 \pm 0.24$ | $7.51 \pm 0.02$ | $19.68 \pm 0.46$ |
| 3 | 1 | GMOCP | $89.60 \pm 0.25$ | $25.23 \pm 2.45$ | $7.29 \pm 0.06$ | $8.37 \pm 0.99$ |
| | | EGMOCP | $89.63 \pm 0.24$ | $10.10 \pm 0.30$ | $10.25 \pm 0.03$ | $22.10 \pm 0.79$ |
| | 2 | GMOCP | $89.60 \pm 0.39$ | $22.35 \pm 1.05$ | $7.75 \pm 0.06$ | $11.08 \pm 0.70$ |
| | | EGMOCP | $89.74 \pm 0.23$ | $9.27 \pm 0.23$ | $10.73 \pm 0.04$ | $23.26 \pm 0.54$ |
| | 4 | GMOCP | $89.64 \pm 0.24$ | $21.91 \pm 0.58$ | $8.47 \pm 0.05$ | $11.19 \pm 0.62$ |
| | | EGMOCP | $89.63 \pm 0.33$ | $8.47 \pm 0.21$ | $11.43 \pm 0.05$ | $23.59 \pm 0.61$ |
| 5 | 1 | GMOCP | $89.54 \pm 0.30$ | $24.08 \pm 1.00$ | $8.52 \pm 0.09$ | $9.34 \pm 1.34$ |
| | | EGMOCP | $89.72 \pm 0.38$ | $8.95 \pm 0.25$ | $12.91 \pm 0.06$ | $23.97 \pm 0.58$ |
| | 2 | GMOCP | $89.75 \pm 0.38$ | $23.87 \pm 1.36$ | $9.09 \pm 0.05$ | $9.95 \pm 0.74$ |
| | | EGMOCP | $89.63 \pm 0.24$ | $8.17 \pm 0.15$ | $13.01 \pm 0.12$ | $24.58 \pm 0.42$ |
| | 4 | GMOCP | $89.83 \pm 0.46$ | $23.66 \pm 0.78$ | $11.13 \pm 0.08$ | $10.53 \pm 0.42$ |
| | | EGMOCP | $89.97 \pm 0.16$ | $7.82 \pm 0.23$ | $15.73 \pm 0.09$ | $24.93 \pm 0.47$ |
| 7 | 1 | GMOCP | $89.79 \pm 0.29$ | $22.72 \pm 2.99$ | $9.18 \pm 0.41$ | $10.36 \pm 1.82$ |
| | | EGMOCP | $89.68 \pm 0.29$ | $8.19 \pm 0.18$ | $13.73 \pm 0.14$ | $24.99 \pm 0.44$ |
| | 2 | GMOCP | $89.70 \pm 0.28$ | $22.52 \pm 1.50$ | $10.32 \pm 0.09$ | $10.64 \pm 0.66$ |
| | | EGMOCP | $89.75 \pm 0.35$ | $7.60 \pm 0.13$ | $14.92 \pm 0.09$ | $25.66 \pm 0.38$ |
| | 4 | GMOCP | $89.74 \pm 0.16$ | $23.72 \pm 0.81$ | $12.04 \pm 0.05$ | $10.40 \pm 0.83$ |
| | | EGMOCP | $89.73 \pm 0.23$ | $7.30 \pm 0.17$ | $16.49 \pm 0.06$ | $25.81 \pm 0.40$ |
| 9 | 1 | GMOCP | $89.81 \pm 0.29$ | $23.09 \pm 2.62$ | $9.79 \pm 0.17$ | $10.34 \pm 1.26$ |
| | | EGMOCP | $89.73 \pm 0.29$ | $7.52 \pm 0.10$ | $14.41 \pm 0.22$ | $25.97 \pm 0.46$ |
| | 2 | GMOCP | $89.69 \pm 0.30$ | $25.23 \pm 2.44$ | $11.35 \pm 0.17$ | $8.70 \pm 1.53$ |
| | | EGMOCP | $89.72 \pm 0.35$ | $7.25 \pm 0.19$ | $16.27 \pm 0.10$ | $26.07 \pm 0.43$ |
| | 4 | GMOCP | $89.66 \pm 0.33$ | $24.26 \pm 0.95$ | $13.47 \pm 0.10$ | $9.99 \pm 0.69$ |
| | | EGMOCP | $89.62 \pm 0.30$ | $\mathbf{7.01 \pm 0.15}$ | $18.12 \pm 0.08$ | $26.41 \pm 0.52$ |

images across 20 classeswhich creates distribution shifts using two distinct transformation sequences. From each sequence, two datasets are generated with random variations to ensure uniqueness across samples. Each dataset contains 3,000 images across 20 classes. Gradual shifts are simulated by sampling within a single transformation type, while sudden shifts are modeled by alternating between datasets from different transformations. Results are presented in Table 6. It can be observed that the two algorithms proposed in this work, GMOCP and EGMOCP, achieve smaller prediction sets compared to the benchmarks across all settings. Please note that, in this set of experiments, there are no prediction sets of size one that cover the true label; therefore, this metric is equal to zero for every setting.

## B.5 Comparison with single model based methods

In this subsection, we compare multi-model based methods with recent adaptive conformal prediction methods designed for online environments. We include several single-model based methods in the comparison, such as ACI (Gibbs & Candès, 2021), FACI (Gibbs & Candès, 2024), DECAY (Angelopoulos et al., 2024), and SAOCP (Bhatnagar et al., 2023). To distinguish between different configurations of single-model methods, we use specific suffixes. For instance, ACI-120D refers to the version using high-performance models, ACI-10R denotes a medium-performance model, and ACI-1R represents a low-performing model. As observed from the table 7, both GMOCP and EGMOCP outperform the medium-performance single-model baselines in terms of average width and single-width ratio. Moreover, EGMOCP produces more efficient prediction sets—smaller average widths and higher single-width ratios—even compared to the high-performance configurations of all single-model methods.

Table 4: Results on the CIFAR-10C dataset under gradual distribution shifts, evaluated across different values of $N$ and $J$. The target coverage is 90%. Bold numbers indicate the best performance in each column. GMOCP consistently achieves faster runtime compared to MOCP across all settings. EGMOCP constructs smaller prediction sets and a higher proportion of single-width sets.

| $N$ | $J$ | Method | Coverage (%) | Avg Width | Run Time | Single Width |
|---|---|---|---|---|---|---|
| | | MOCP | **90.03 $\pm$ 0.30** | 2.07 $\pm$ 0.35 | 9.41 $\pm$ 0.08 | 48.00 $\pm$ 7.84 |
| | | COMA | 90.00 $\pm$ 0.02 | 1.49 $\pm$ 0.07 | 11.02 $\pm$ 0.03 | **61.39 $\pm$ 2.92** |
| 1 | 1 | GMOCP | 89.36 $\pm$ 0.21 | 1.90 $\pm$ 0.27 | **4.29 $\pm$ 0.02** | 48.14 $\pm$ 4.22 |
| | | EGMOCP | 89.37 $\pm$ 0.22 | 1.52 $\pm$ 0.03 | 5.47 $\pm$ 0.02 | 57.48 $\pm$ 1.58 |
| | 2 | GMOCP | 89.39 $\pm$ 0.29 | 1.79 $\pm$ 0.03 | 4.48 $\pm$ 0.02 | 52.06 $\pm$ 0.72 |
| | | EGMOCP | 89.40 $\pm$ 0.23 | 1.57 $\pm$ 0.02 | 5.68 $\pm$ 0.12 | 55.95 $\pm$ 1.14 |
| | 4 | GMOCP | 89.26 $\pm$ 0.15 | 1.76 $\pm$ 0.04 | 4.85 $\pm$ 0.05 | 52.78 $\pm$ 0.90 |
| | | EGMOCP | 89.27 $\pm$ 0.12 | 1.55 $\pm$ 0.02 | 6.13 $\pm$ 0.08 | 56.74 $\pm$ 0.96 |
| 3 | 1 | GMOCP | 89.79 $\pm$ 0.25 | 1.78 $\pm$ 0.17 | 5.51 $\pm$ 0.06 | 50.97 $\pm$ 3.41 |
| | | EGMOCP | 89.83 $\pm$ 0.30 | 1.50 $\pm$ 0.02 | 8.61 $\pm$ 0.02 | 58.98 $\pm$ 1.07 |
| | 2 | GMOCP | 89.98 $\pm$ 026 | 1.78 $\pm$ 0.04 | 6.00 $\pm$ 0.04 | 53.03 $\pm$ 1.37 |
| | | EGMOCP | 89.76 $\pm$ 0.36 | 1.48 $\pm$ 0.01 | 9.04 $\pm$ 0.03 | 59.13 $\pm$ 0.86 |
| | 4 | GMOCP | 89.69 $\pm$ 0.28 | 1.70 $\pm$ 0.04 | 6.80 $\pm$ 0.05 | 54.26 $\pm$ 1.25 |
| | | EGMOCP | 89.53 $\pm$ 0.28 | 1.44 $\pm$ 0.02 | 9.85 $\pm$ 0.02 | 60.17 $\pm$ 0.98 |
| 5 | 1 | GMOCP | 89.72 $\pm$ 0.17 | 1.85 $\pm$ 0.15 | 6.35 $\pm$ 0.12 | 51.02 $\pm$ 2.96 |
| | | EGMOCP | 89.82 $\pm$ 0.26 | 1.48 $\pm$ 0.02 | 10.91 $\pm$ 0.03 | 59.39 $\pm$ 0.83 |
| | 2 | GMOCP | 89.80 $\pm$ 0.35 | 1.78 $\pm$ 0.06 | 7.24 $\pm$ 0.05 | 52.62 $\pm$ 1.72 |
| | | EGMOCP | 89.87 $\pm$ 0.18 | 1.44 $\pm$ 0.01 | 11.65 $\pm$ 0.03 | 60.30 $\pm$ 0.75 |
| | 4 | GMOCP | 89.82 $\pm$ 0.23 | 1.80 $\pm$ 0.03 | 8.47 $\pm$ 0.06 | 52.44 $\pm$ 0.88 |
| | | EGMOCP | 89.97 $\pm$ 0.28 | **1.43 $\pm$ 0.01** | 12.86 $\pm$ 0.04 | 60.67 $\pm$ 0.59 |

Table 5: Results on the TinyImageNet dataset under gradual distribution shifts, evaluated across different values of $N$ and $J$. The target coverage is 90%. Bold numbers denote the best results in each column. Bold numbers indicate the best performance in each column. GMOCP consistently achieves faster runtime compared to MOCP across all settings. EGMOCP constructs smaller prediction sets.

| $N$ | $J$ | Method | Coverage (%) | Avg Width | Run Time | Width < 40 |
|---|---|---|---|---|---|---|
| | | MOCP | **89.61 $\pm$ 0.46** | 170.59 $\pm$ 1.03 | 3.01 $\pm$ 0.01 | 0.11 $\pm$ 0.05 |
| | | COMA | 90.00 $\pm$ 0.05 | 161.61 $\pm$ 1.56 | 3.91 $\pm$ 0.01 | **0.34 $\pm$ 0.12** |
| 1 | 1 | GMOCP | 87.90 $\pm$ 0.28 | 165.68 $\pm$ 1.34 | **2.03 $\pm$ 0.01** | 0.12 $\pm$ 0.09 |
| | | EGMOCP | 87.67 $\pm$ 0.35 | 164.51 $\pm$ 1.16 | 2.34 $\pm$ 0.01 | 0.15 $\pm$ 0.06 |
| | 2 | GMOCP | 87.81 $\pm$ 0.40 | 165.57 $\pm$ 2.08 | 2.12 $\pm$ 0.03 | 0.20 $\pm$ 0.18 |
| | | EGMOCP | 87.95 $\pm$ 0.36 | 165.43 $\pm$ 1.33 | 2.44 $\pm$ 0.04 | 0.20 $\pm$ 0.10 |
| | 4 | GMOCP | 87.54 $\pm$ 0.27 | 165.28 $\pm$ 1.77 | 2.26 $\pm$ 0.01 | 0.16 $\pm$ 0.11 |
| | | EGMOCP | 87.65 $\pm$ 0.33 | **164.28 $\pm$ 1.75** | 2.58 $\pm$ 0.00 | 0.18 $\pm$ 0.17 |
| 3 | 1 | GMOCP | 88.91 $\pm$ 0.37 | 167.66 $\pm$ 1.43 | 2.38 $\pm$ 0.02 | 0.11 $\pm$ 0.05 |
| | | EGMOCP | 88.86 $\pm$ 0.42 | 166.97 $\pm$ 1.73 | 3.12 $\pm$ 0.01 | 0.16 $\pm$ 0.09 |
| | 2 | GMOCP | 88.98 $\pm$ 0.38 | 168.01 $\pm$ 1.29 | 2.56 $\pm$ 0.03 | 0.08 $\pm$ 0.06 |
| | | EGMOCP | 89.04 $\pm$ 0.53 | 167.07 $\pm$ 1.50 | 3.34 $\pm$ 0.05 | 0.19 $\pm$ 0.12 |
| | 4 | GMOCP | 88.98 $\pm$ 0.34 | 168.98 $\pm$ 1.01 | 2.85 $\pm$ 0.02 | 0.12 $\pm$ 0.07 |
| | | EGMOCP | 89.14 $\pm$ 0.50 | 167.24 $\pm$ 1.41 | 3.62 $\pm$ 0.01 | 0.18 $\pm$ 0.11 |

## B.6 Candidate models with little variance analysis

In this subsection, we provide additional experiments to evaluate how the proposed methods perform when all candidate models have similar quality. To do this, we select six high-performance models introduced in Section 4. As shown in Table 8, GMOCP reduces the computational complexity, while EGMOCP still produces efficient prediction sets with smaller average widths and higher single-width ratios, even when there is little variance among the models.

Additionally, we note that "Variance introduced by model selection" and "instability under distribution shift" can impact coverage. By comparing Table 8 and Table 1, it can be observed that changing the distribution shift from sudden to gradual and employing candidate models with low variance in quality results in improved coverage. For example, for EGMOCP with $(J = 4, n = 1)$, the coverage improved from 88.99 (Table 1) to 89.43 (Table 8).

Table 6: Results on the Synthetic dataset, evaluated across different values of $N$ and $J$. The target coverage is 90%. Bold numbers denote the best results in each column. GMOCP consistently achieves faster runtime compared to MOCP across all settings. EGMOCP constructs smaller prediction sets and a higher proportion of single-width sets.

| $N$ | $J$ | Method | Coverage (%) | Avg Width | Run Time | Single Width |
|-----|-----|--------|--------------|-----------|----------|--------------|
|   |   | MOCP | $89.92 \pm 0.28$ | $18.00 \pm 0.01$ | $13.57 \pm 0.18$ | $0.00 \pm 0.00$ |
|   |   | COMA | $89.99 \pm 0.01$ | $18.01 \pm 0.04$ | $16.09 \pm 0.05$ | $0.00 \pm 0.00$ |
| 1 | 1 | GMOCP | $89.24 \pm 0.12$ | $17.88 \pm 0.06$ | $\mathbf{5.96 \pm 0.05}$ | $0.00 \pm 0.00$ |
|   |   | EGMOCP | $89.18 \pm 0.15$ | $17.84 \pm 0.12$ | $22.44 \pm 0.39$ | $0.00 \pm 0.00$ |
|   | 2 | GMOCP | $89.26 \pm 0.15$ | $17.86 \pm 0.07$ | $6.17 \pm 0.05$ | $0.00 \pm 0.00$ |
|   |   | EGMOCP | $89.27 \pm 0.19$ | $17.85 \pm 0.07$ | $22.70 \pm 0.44$ | $0.00 \pm 0.00$ |
| 3 | 1 | GMOCP | $89.60 \pm 0.20$ | $17.96 \pm 0.04$ | $7.17 \pm 0.27$ | $0.00 \pm 0.00$ |
|   |   | EGMOCP | $89.53 \pm 0.20$ | $17.92 \pm 0.04$ | $27.96 \pm 0.49$ | $0.00 \pm 0.00$ |
|   | 2 | GMOCP | $89.60 \pm 0.17$ | $17.96 \pm 0.02$ | $7.76 \pm 0.08$ | $0.00 \pm 0.00$ |
|   |   | EGMOCP | $89.47 \pm 0.22$ | $17.93 \pm 0.04$ | $28.37 \pm 0.51$ | $0.00 \pm 0.00$ |
| 5 | 1 | GMOCP | $89.77 \pm 0.27$ | $17.95 \pm 0.02$ | $7.88 \pm 0.45$ | $0.00 \pm 0.00$ |
|   |   | EGMOCP | $89.70 \pm 0.28$ | $\mathbf{17.94 \pm 0.02}$ | $31.90 \pm 0.56$ | $0.00 \pm 0.00$ |
|   | 2 | GMOCP | $89.73 \pm 0.26$ | $17.97 \pm 0.04$ | $8.90 \pm 0.14$ | $0.00 \pm 0.00$ |
|   |   | EGMOCP | $\mathbf{89.66 \pm 0.41}$ | $17.94 \pm 0.03$ | $32.90 \pm 0.60$ | $0.00 \pm 0.00$ |

Table 7: Results on the CIFAR-100C dataset under sudden distribution shifts, comparison of GMOCP($N = 3, J = 1$) and EGMOCP ($N = 5, J = 4$) single model based algorithms. EGMOCP yields more efficient prediction sets—smaller average widths and higher single-width ratios— even compared to high-performance single-model baselines.

| Method | Coverage (%) | Avg Width | Run Time | Single Width |
|--------|--------------|-----------|----------|--------------|
| GMOCP | $89.55 \pm 0.28$ | $10.68 \pm 1.05$ | $6.05 \pm 0.04$ | $23.45 \pm 2.53$ |
| EGMOCP | $89.43 \pm 0.27$ | $\mathbf{6.18 \pm 0.14}$ | $12.97 \pm 0.05$ | $\mathbf{29.91 \pm 0.47}$ |
| MOCP | $89.71 \pm 0.37$ | $12.63 \pm 3.53$ | $9.37 \pm 0.05$ | $22.43 \pm 2.53$ |
| COMA | $\mathbf{90.00 \pm 0.01}$ | $8.36 \pm 0.95$ | $11.23 \pm 0.07$ | $28.60 \pm 1.83$ |
| FACI-120D | $89.66 \pm 0.36$ | $6.97 \pm 1.07$ | $4.49 \pm 0.02$ | $27.82 \pm 0.91$ |
| FACI-10R | $89.64 \pm 0.34$ | $12.88 \pm 1.85$ | $4.47 \pm 0.03$ | $5.30 \pm 0.51$ |
| FACI-1R | $89.75 \pm 0.29$ | $49.81 \pm 0.65$ | $4.48 \pm 0.04$ | $0.00 \pm 0.01$ |
| ACI-120D | $89.96 \pm 0.02$ | $9.55 \pm 0.54$ | $3.14 \pm 0.01$ | $26.08 \pm 0.51$ |
| ACI-10R | $89.95 \pm 0.02$ | $15.52 \pm 0.69$ | $3.16 \pm 0.03$ | $5.43 \pm 0.46$ |
| ACI-1R | $89.96 \pm 0.03$ | $51.76 \pm 0.72$ | $3.15 \pm 0.02$ | $0.00 \pm 0.00$ |
| DECAY-120D | $89.76 \pm 0.01$ | $6.42 \pm 0.10$ | $1.41 \pm 0.01$ | $28.07 \pm 0.53$ |
| DECAY-10R | $89.75 \pm 0.10$ | $12.12 \pm 0.33$ | $\mathbf{1.40 \pm 0.02}$ | $4.91 \pm 0.51$ |
| DECAY-1R | $89.80 \pm 0.08$ | $49.72 \pm 0.61$ | $1.41 \pm 0.02$ | $0.01 \pm 0.01$ |
| SAOCP-120D | $88.65 \pm 0.15$ | $7.26 \pm 0.18$ | $25.52 \pm 0.03$ | $26.99 \pm 0.66$ |
| SAOCP-10R | $88.50 \pm 0.18$ | $13.27 \pm 0.38$ | $25.52 \pm 0.07$ | $6.60 \pm 0.54$ |
| SAOCP-1R | $88.29 \pm 0.10$ | $48.74 \pm 0.79$ | $25.51 \pm 0.06$ | $0.00 \pm 0.00$ |

## B.7 Local coverage comparison

In this subsection, we present a new set of experiments to evaluate the empirical coverage of our algorithms over small windows, rather than across the entire time horizon. Specifically, we report the local coverage using a window size of 100 in Table 9. As shown, both of our proposed algorithms, GMOCP and EGMOCP, achieve local coverage close to the target level. This result validates the ability of our methods to quickly adapt to abrupt distribution shifts.

## B.8 Comparison of set size at each time step

We conducted an experiment to compare the prediction set sizes generated by each algorithm over the full time horizon (6000 data points). As shown in Figure 3, EGMOCP achieves the smallest average prediction

Table 8: Results on the CIFAR-100C dataset under gradual distribution shifts, where candidate models have little variance in quality. EGMOCP yields more efficient prediction sets, smaller average widths and higher single-width ratios.

| N | J | Method | Coverage (%) | Avg Width | Run Time | Single Width |
|---|---|---|---|---|---|---|
|  |  | MOCP | 89.89 ± 0.23 | 5.63 ± 0.15 | 7.97 ± 0.03 | 29.66 ± 0.85 |
|  |  | COMA | **90.00 ± 0.02** | 7.21 ± 0.31 | 9.05 ± 0.08 | 28.84 ± 1.67 |
| 1 | 2 | GMOCP | 89.49 ± 0.13 | 5.62 ± 0.13 | **4.53 ± 0.02** | 29.83 ± 0.54 |
|  |  | EGMOCP | 89.46 ± 0.15 | 5.57 ± 0.14 | 5.69 ± 0.02 | 30.01 ± 0.79 |
|  | 4 | GMOCP | 89.46 ± 0.25 | 5.61 ± 0.17 | 4.71 ± 0.05 | 29.85 ± 0.63 |
|  |  | EGMOCP | 89.43 ± 0.19 | 5.61 ± 0.14 | 5.80 ± 0.02 | 30.05 ± 0.64 |
| 3 | 2 | GMOCP | 89.86 ± 0.23 | 5.66 ± 0.14 | 5.99 ± 0.03 | 29.66 ± 0.68 |
|  |  | EGMOCP | 89.76 ± 0.22 | **5.55 ± 0.13** | 8.76 ± 0.04 | **30.28 ± 0.63** |
|  | 4 | GMOCP | 89.99 ± 0.21 | 5.62 ± 0.15 | 6.71 ± 0.05 | 29.88 ± 0.77 |
|  |  | EGMOCP | 89.81 ± 0.25 | 5.58 ± 0.19 | 9.49 ± 0.03 | 30.13 ± 1.03 |

Table 9: Results on the CIFAR-100C dataset under sudden distribution shifts, showing empirical coverage over local windows (window size = 100). Both GMOCP and EGMOCP maintain local coverage close to the desired level of %90

| N | J | Method | Coverage (%) | Local Coverage (%) |
|---|---|---|---|---|
|  |  | MOCP | 89.71 ± 0.37 | 89.79 ± 0.38 |
|  |  | COMA | 90.00 ± 0.01 | 89.99 ± 0.01 |
| 1 | 1 | GMOCP | 89.11 ± 0.21 | 89.14 ± 0.24 |
|  |  | EGMOCP | 89.10 ± 0.28 | 89.14 ± 0.30 |
|  | 2 | GMOCP | 89.10 ± 0.19 | 89.14 ± 0.21 |
|  |  | EGMOCP | 89.03 ± 0.17 | 89.09 ± 0.18 |
|  | 4 | GMOCP | 89.04 ± 0.21 | 89.08 ± 0.23 |
|  |  | EGMOCP | 88.99 ± 0.21 | 89.04 ± 0.23 |
| 3 | 1 | GMOCP | 89.55 ± 0.28 | 89.60 ± 0.29 |
|  |  | EGMOCP | 89.38 ± 0.22 | 89.46 ± 0.22 |
|  | 2 | GMOCP | 89.50 ± 0.26 | 89.59 ± 0.25 |
|  |  | EGMOCP | 89.29 ± 0.21 | 89.36 ± 0.23 |
|  | 4 | GMOCP | 89.64 ± 0.34 | 89.72 ± 0.33 |
|  |  | EGMOCP | 89.38 ± 0.31 | 89.45 ± 0.32 |
| 5 | 1 | GMOCP | 89.55 ± 0.30 | 89.64 ± 0.32 |
|  |  | EGMOCP | 89.46 ± 0.36 | 89.54 ± 0.37 |
|  | 2 | GMOCP | 89.53 ± 0.26 | 89.61 ± 0.28 |
|  |  | EGMOCP | 89.44 ± 0.28 | 89.50 ± 0.29 |
|  | 4 | GMOCP | 89.73 ± 0.31 | 89.82 ± 0.33 |
|  |  | EGMOCP | 89.43 ± 0.27 | 89.49 ± 0.28 |

set size among all benchmarks. COMA exhibits significant fluctuations, resulting in large prediction sets at certain timesteps.

## B.9 Experimental results for CIFAR10-C

We conducted experiments on the CIFAR-10C dataset under two different settings. Table 10 corresponds to the sudden distribution shift scenario with equal exploration ratios across selective nodes. Table 11 presents results for the gradual distribution shift scenario with unequal exploration ratios, as detailed in Section 4.

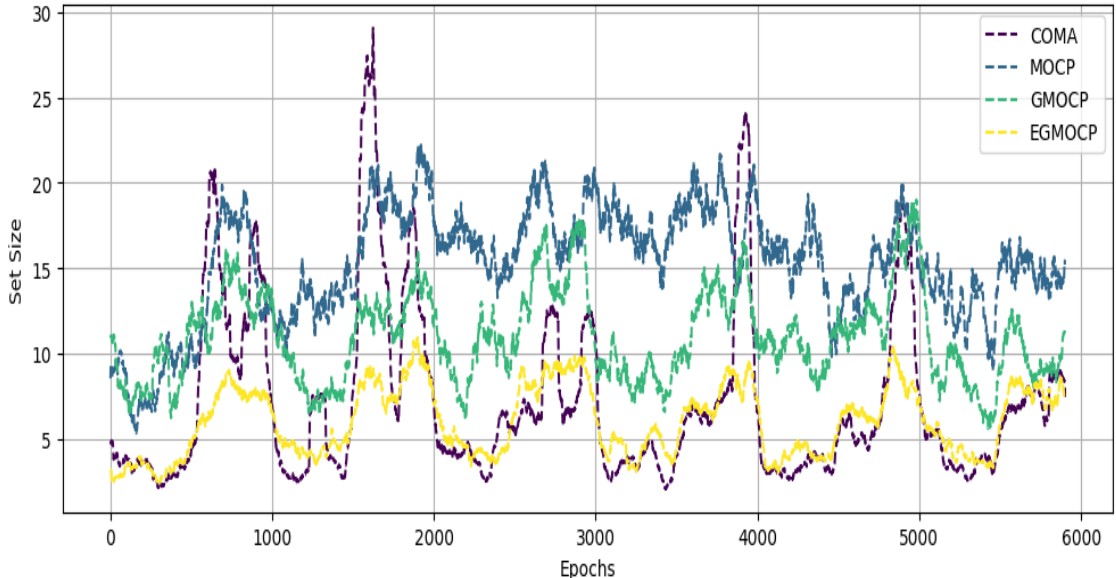

Figure 3: Evaluation of prediction set sizes constructed by multi-model methods over 6000 timesteps. On average, EGMOCP produces smaller prediction sets compared to all other benchmarks. The average prediction set sizes for COMA, MOCP, GMOCP, and EGMOCP across the 6000 timesteps are 7.19, 15.26, 11.00, and 6.06, respectively.

Table 10: Results on the CIFAR-10C dataset under sudden distribution shifts, evaluated across different values of $N$ and $J$. The target coverage is 90%. Bold numbers denote the best results in each column. GMOCP consistently achieves faster runtime compared to MOCP across all settings. EGMOCP constructs smaller prediction sets and a higher proportion of single-width sets.

| $N$ | $J$ | Method | Coverage (%) | Avg Width | Run Time | Single Width |
|---|---|---|---|---|---|---|
| | | MOCP | $89.63 \pm 0.26$ | $1.94 \pm 0.30$ | $9.49 \pm 0.06$ | $50.04 \pm 6.54$ |
| | | COMA | $\mathbf{89.99 \pm 0.01}$ | $1.65 \pm 0.10$ | $10.89 \pm 0.02$ | $57.56 \pm 2.35$ |
| 1 | 1 | GMOCP | $89.09 \pm 0.16$ | $1.81 \pm 0.13$ | $\mathbf{4.31 \pm 0.03}$ | $50.65 \pm 2.35$ |
| | | EGMOCP | $88.99 \pm 0.17$ | $1.53 \pm 0.03$ | $5.51 \pm 0.02$ | $57.04 \pm 1.41$ |
| | 2 | GMOCP | $89.09 \pm 0.30$ | $1.81 \pm 0.03$ | $4.46 \pm 0.02$ | $51.96 \pm 0.95$ |
| | | EGMOCP | $89.12 \pm 0.28$ | $1.58 \pm 0.02$ | $5.67 \pm 0.01$ | $56.43 \pm 1.13$ |
| | 4 | GMOCP | $88.89 \pm 0.21$ | $1.75 \pm 0.02$ | $4.95 \pm 0.04$ | $53.39 \pm 1.03$ |
| | | EGMOCP | $88.91 \pm 0.18$ | $1.54 \pm 0.02$ | $6.22 \pm 0.09$ | $57.01 \pm 1.20$ |
| 3 | 1 | GMOCP | $89.56 \pm 0.29$ | $1.74 \pm 0.16$ | $5.41 \pm 0.11$ | $52.03 \pm 3.45$ |
| | | EGMOCP | $89.52 \pm 0.30$ | $1.50 \pm 0.02$ | $8.49 \pm 0.02$ | $59.24 \pm 1.18$ |
| | 2 | GMOCP | $89.52 \pm 0.32$ | $1.76 \pm 0.04$ | $5.98 \pm 0.05$ | $53.27 \pm 0.72$ |
| | | EGMOCP | $89.35 \pm 0.31$ | $1.49 \pm 0.01$ | $9.00 \pm 0.04$ | $59.01 \pm 0.93$ |
| | 4 | GMOCP | $89.57 \pm 0.36$ | $1.71 \pm 0.03$ | $6.33 \pm 0.08$ | $53.94 \pm 1.26$ |
| | | EGMOCP | $89.36 \pm 0.17$ | $1.45 \pm 0.02$ | $9.72 \pm 0.04$ | $59.97 \pm 0.89$ |
| 5 | 1 | GMOCP | $89.72 \pm 0.24$ | $1.82 \pm 0.22$ | $6.72 \pm 0.07$ | $52.13 \pm 5.11$ |
| | | EGMOCP | $89.53 \pm 0.23$ | $1.49 \pm 0.01$ | $10.80 \pm 0.04$ | $59.43 \pm 0.76$ |
| | 2 | GMOCP | $89.59 \pm 0.24$ | $1.79 \pm 0.05$ | $7.22 \pm 0.07$ | $52.32 \pm 0.95$ |
| | | EGMOCP | $89.58 \pm 0.24$ | $1.45 \pm 0.02$ | $11.62 \pm 0.03$ | $\mathbf{60.13 \pm 1.03}$ |
| | 4 | GMOCP | $89.67 \pm 0.25$ | $1.81 \pm 0.04$ | $8.31 \pm 0.04$ | $52.64 \pm 0.65$ |
| | | EGMOCP | $89.55 \pm 0.27$ | $\mathbf{1.44 \pm 0.01}$ | $12.65 \pm 0.04$ | $60.00 \pm 0.53$ |

Table 11: Results on the CIFAR-10C dataset under gradual distribution shifts, evaluated across different values of $N$ and $J$. The target coverage is 90%. Bold numbers denote the best results in each column. GMOCP consistently achieves faster runtime compared to MOCP across all settings. EGMOCP constructs smaller prediction sets and a higher proportion of single-width sets.

| $N$ | $J$ | Method | Coverage (%) | Avg Width | Run Time | Single Width |
|---|---|---|---|---|---|---|
| | | MOCP | **90.03** ± 0.30 | 2.07 ± 0.35 | 8.83 ± 0.04 | 48.00 ± 7.84 |
| | | COMA | 90.02 ± 0.02 | 1.49 ± 0.07 | 10.76 ± 0.04 | 61.39 ± 2.92 |
| 1 | 1 | GMOCP | 89.36 ± 0.21 | 1.90 ± 0.27 | **4.32 ± 0.01** | 48.14 ± 4.22 |
| | | EGMOCP | 89.37 ± 0.22 | 1.52 ± 0.03 | 5.51 ± 0.06 | 57.48 ± 1.58 |
| | 2 | GMOCP | 89.37 ± 0.17 | 1.78 ± 0.03 | 4.39 ± 0.03 | 52.30 ± 1.27 |
| | | EGMOCP | 89.35 ± 0.21 | 1.59 ± 0.02 | 5.58 ± 0.01 | 55.57 ± 1.19 |
| | 4 | GMOCP | 89.25 ± 0.21 | 1.77 ± 0.04 | 4.74 ± 0.03 | 52.45 ± 1.03 |
| | | EGMOCP | 89.28 ± 0.16 | 1.55 ± 0.02 | 5.91 ± 0.01 | 56.69 ± 1.11 |
| 3 | 1 | GMOCP | 89.79 ± 0.25 | 1.78 ± 0.17 | 5.41 ± 0.06 | 50.97 ± 3.41 |
| | | EGMOCP | 89.83 ± 0.30 | 1.50 ± 0.02 | 8.44 ± 0.03 | 58.98 ± 1.07 |
| | 2 | GMOCP | 89.78 ± 0.32 | 1.78 ± 0.04 | 6.06 ± 0.05 | 52.64 ± 1.23 |
| | | EGMOCP | 89.65 ± 0.22 | 1.37 ± 0.01 | 9.08 ± 0.05 | 61.51 ± 0.77 |
| | 4 | GMOCP | 89.72 ± 0.28 | 1.73 ± 0.03 | 6.36 ± 0.04 | 53.68 ± 1.17 |
| | | EGMOCP | 89.60 ± 0.35 | 1.41 ± 0.02 | 9.73 ± 0.02 | 60.68 ± 1.21 |
| 5 | 1 | GMOCP | 89.72 ± 0.17 | 1.85 ± 0.15 | 6.71 ± 0.11 | 51.02 ± 2.96 |
| | | EGMOCP | 89.82 ± 0.26 | 1.48 ± 0.02 | 10.88 ± 0.04 | 59.39 ± 0.83 |
| | 2 | GMOCP | 89.73 ± 0.26 | 1.80 ± 0.03 | 7.33 ± 0.03 | 52.27 ± 1.13 |
| | | EGMOCP | 89.87 ± 0.36 | **1.35 ± 0.01** | 11.66 ± 0.03 | **62.23 ± 0.57** |
| | 4 | GMOCP | 89.88 ± 0.21 | 1.80 ± 0.02 | 8.47 ± 0.07 | 52.49 ± 0.92 |
| | | EGMOCP | 89.99 ± 0.24 | 1.38 ± 0.02 | 12.84 ± 0.11 | 61.84 ± 0.55 |

