# OpenReview forum: "Multi-model Online Conformal Prediction with Graph-Structured Feedback"
_TMLR — Accepted by TMLR_

### Review · Reviewer_djnH · 2025-07-08

**Summary Of Contributions:**

This paper introduces two novel online conformal prediction algorithms: Graph-structured feedback Multimodel Ensemble Online Conformal Prediction (GMOCP) and its extension, Efficient GMOCP (EGMOCP). In particular, GMOCP employs a bipartite graph structure to identify a subset of effective models at each time step, aiming to reduce computational cost and avoid using poor-performing models. EGMOCP enhances GMOCP by incorporating the prediction set size into the model selection process, resulting in small and informative prediction sets. The authors provide regret and coverage bounds for both methods, demonstrating sublinear regret and bounded deviation from target coverage. Experiments on CIFAR-10C and CIFAR-100C under both sudden and gradual distribution shifts show that GMOCP and EGMOCP achieve smaller prediction sets and lower runtime than several multi-model conformal prediction baselines.

**Audience:**

Yes

**Claims And Evidence:**

No

**Requested Changes:**

I remain open to being convinced if certain suggestions are not feasible or appropriate.

1. A theoretical analysis of both computational complexity and prediction efficiency should be provided to substantiate the corresponding claims. This could involve comparing the proposed methods against full-model ensembles such as MOCP.
2. The experimental evaluation should include additional baseline methods such as ACI and SOACP, to support the claimed improvements in computational complexity and prediction efficiency.
3. The paper should offer a reasonable explanation for the observed coverage deviations of GMOCP and EGMOCP from the target level, such as variance introduced by model selection, instability under distribution shift. Additionally, I encourage the authors to discuss or propose strategies for mitigating these deviations.
4. While the paper highlights that multi-model online conformal prediction methods can suffer due to the poor performance of some candidate models, it remains unclear how the proposed approach behaves when all candidate models perform similarly. I would like to see additional analysis or experiments examining the method's effectiveness in such scenarios, where there is little variance in model quality.
5. The paper could benefit from an evaluation of the local performance of the proposed methods. One notable advantage of multi-model approaches is their ability to quickly adapt to dynamic environments, potentially achieving local coverage that is close to the target level of $1-\alpha$ [1]. It would be insightful to see whether the proposed methods, GMOCP and EGMOCP, can achieve comparable local coverage performance to the baseline methods, especially in situations where models are exposed to rapidly changing distributions.
6. The introduction currently provides an incomplete and somewhat high-level overview of online conformal prediction (OCP), which is difficult to follow for readers who are not already familiar with the topic. I recommend that the authors offer a more comprehensive introduction to the OCP framework.

### References
[1] Bhatnagar A, et al. Improved online conformal prediction via strongly adaptive online learning. ICML 2023.

**Strengths And Weaknesses:**

### Strengths
This paper introduces two algorithms, GMOCP and EGMOCP, for online conformal prediction. The key innovation lies in dynamically selecting a subset of models at each timestep. To the best of my knowledge, this is the first work to integrate graph-based combinatorial selection with conformal prediction in an online setting, making it a novel and well-motivated contribution. The authors support their approach with rigorous theoretical analysis, establishing valid marginal coverage guarantees and sublinear regret bounds, both of which are standard desiderata in the online conformal prediction literature. The empirical evaluation is conducted across multiple datasets (CIFAR-10C and CIFAR-100C), corruption types, and model architectures.  Results consistently show that GMOCP and EGMOCP outperform existing baselines such as MOCP and COMA in terms of average prediction set width and computational efficiency. Overall, this work represents a meaningful advancement in the field of online conformal prediction and is likely to be of strong interest to the TMLR community, particularly to researchers working on conformal prediciton.

### Weaknesses
In my opinion, some of the claims in this paper is not well-supported. In particular, the authors propose to select a subset of effective models and incorporate the prediction set size as feedback, claiming that this reduces computational complexity and improves prediction efficiency. While this is a compelling intuition, the paper lacks a formal analysis or theoretical justification of these benefits. For instance, there is no analysis of prediction set efficiency comparing the proposed methods to existing ensemble conformal prediction baselines. Given the theoretical emphasis in the conformal prediction community, such guarantees are important and should be provided.

Besides, the experiments results lack comparison with several important baseline methods such as standard adaptive conformal inference (ACI) with both constant learning rate [1] and dynamic learning rates [2] and Strongly Adaptive Online Conformal Prediction (SOACP) [3]. These methods, though are not multi-model algorithms, are important baselines in online conformal prediction. This lack of comprehensive baselines weakens the validity of the empirical superiority in a broader extent.

In addition, the empirical results raise concerns about the coverage validity of the proposed methods. A central claim of the paper is that both GMOCP and EGMOCP maintain valid coverage guarantees. In Tables 1 and 2, GMOCP and EGMOCP often produce coverage rates that deviate noticeably from the target level of 90%, unlike MOCP and COMA. For example, in Table 1, for $N=1$ and $J=4$, EGMOCP reports a coverage of only 88.99 $\pm$ 0.21, which deviates significantly from the target level of 90. These discrepancies are not addressed or explained in the paper, raising questions about the practical reliability of the methods in achieving guaranteed coverage.

### References
[1] Gibbs I, et al. Adaptive conformal inference under distribution shift. NIPS 2021.

[2] Angelopoulos A N, Barber R F, Bates S. Online conformal prediction with decaying step sizes. ICML 2024.

[3] Bhatnagar A, et al. Improved online conformal prediction via strongly adaptive online learning. ICML 2023.

---

> ### Author Response · Authors · 2025-08-09
>
> We want to thank the reviewer for recognizing the novelty and motivation behind our work, and for highlighting it as the first to integrate graph-based structure into conformal prediction in an online setting. We are delighted that you consider our contribution a meaningful advancement in the field. Please find our responses to the weaknesses(W) and Requested Changes(RC) below.
>
> **W1+RC1 Theoretical analysis of both computational complexity and prediction efficiency:** We thank the reviewer for raising the question regarding the theoretical justification of our claims on computational complexity and prediction efficiency. This motivates us to further dive into deeper theoretical justification. Specifically, we provide a theoretical analysis of the efficiency of the prediction sets constructed by GMOCP (our algorithm) compared to COMA. The result is summarized in Lemma 6, and the detailed proof is included in Appendix A.5. It is demonstrated that the upper bound on the expected size of the prediction set constructed by GMOCP at each time $t$ is half of the corresponding upper bound for the COMA algorithm.
>
> Regarding computational complexity, the per-iteration cost at time $t$ for GMOCP is $O(Nt + JMN)$, where the $JMN$ term accounts for graph generation and the $Nt$ term arises from computing $\bar{\alpha}_t^m$ for up to $N$ selected models, using sorted calibration scores of length $t$. The per-iteration complexity of MOCP is $O(Mt)$. Thus, GMOCP can reduce the per-iteration complexity compared to MOCP, especially in settings where $N << M$.
>
> **W2+RC2 Comparison with single-model baselines:** We thank the reviewer for this valuable suggestion, which motivates us to carry out additional experiments to demonstrate that our proposed approach, EGMOCP, results in smaller prediction sets and higher single-width compared to all considered single-model baselines. We have included a comprehensive comparison with all the single-model baselines mentioned by the reviewer: ACI[1], SOACP[2], and DECAY[3]. In addition, we include FACI[4], an enhanced version of ACI that mitigates step-size tuning issues. The results are presented in Appendix B.5. To distinguish between different configurations of single-model methods, we use specific suffixes. For example, ACI-120D refers to a high-performance model, ACI-10R to a medium-performance model, and ACI-1R to a low-performance model. As shown in the results, both GMOCP and EGMOCP outperform the medium-performance single-model baselines in terms of average width and single-width ratio. Moreover, EGMOCP achieves more efficient prediction sets—even compared to high-performance single-model variants—by producing smaller average widths and higher single-width ratios.
>
> **W3+RC3 Deviation from the coverage level:** We thank the reviewer for this accurate question. The observed deviation in empirical coverage for EGMOCP is partially due to the choice of parameters $\eta, \epsilon,$ and $\beta$. In our experiments, we selected these values using a grid search on the CIFAR100-C dataset with a sudden distribution shift, specifically for the GMOCP algorithm with $N=1$ and $J=1$, and then applied the same values across all other experimental settings. However, since the optimal parameters can vary depending on the dataset and the nature of the distribution shift, using a single set of values may have contributed to the coverage deviations noted by the reviewer. To mitigate this, we suggest tuning the parameters separately for each setting. In our experiments, we used a fixed value $\epsilon = 0.5$ across all settings for consistency. We observed that fine-tuning this parameter based on the characteristics of each specific graph can reduce coverage deviation. For instance, as shown in the following table, decreasing $\epsilon$ from 0.5 to 0.4 improves the empirical coverage of EGMOCP $(J=4, N=1)$ with CIFAR-100C under sudden distribution shift.
>
> **Change ε to 0.4 (N1, J4)**
>
> | ε    | Method | Coverage (%)       | Avg Width        | Run Time         | Single Width     |
> |------|--------|--------------------|------------------|------------------|------------------|
> | 0.4  | EGMOCP | 89.01 ± 0.17        | 6.89 ± 0.20       | 11.98 ± 0.06      | 28.77 ± 0.88     |
> | 0.5  | EGMOCP | 88.99 ± 0.21        | 6.92 ± 0.19       | 11.87 ± 0.07      | 28.50 ± 0.80     |

---

> ### Author Response · Authors · 2025-08-09
>
> **W3+RC3 (Continued):** Additionally, we also agree with the reviewer regarding the potential alternative causes of coverage deviation, meaning  ''variance introduced by model selection'' and ''instability under distribution shift.'' We observed that when the employed candidate models have lower variance in quality, the coverage of the proposed algorithm indeed improved. See the response to **RC4**, as well as Table 1 and Table 8 in the revised manuscript. For example, for EGMOCP with $(J=4,n=1)$, the coverage improved from 88.99 (Table 1) to 89.43 (Table 8) when the setting is changed from a sudden to a gradual distribution shift and using high-performance candidate models which corresponds to lower ''variance introduced by model selection '' and lower ''instability under distribution shift.''. This suggests the importance of choosing more reliable and relevant candidate models.  We thank the reviewer for this insightful suggestion. This coverage improvement is discussed in Appendix B.6 of the revised version.
>
>
> **RC4 Additional experiments with little variance in model quality:** To address the reviewer's question about scenarios where all candidate models have similar quality, we have added new experiments in Appendix B.6. Specifically, we select six high-performance models introduced in the Experiments section. It can be observed that GMOCP reduces the computational complexity. EGMOCP still produces efficient prediction sets with smaller average widths and higher single-width ratios, even when there is little variance among the models.
>
> **RC5 Additional experiments for local coverage:** We added a new set of experiments in Appendix B.7 to assess the local coverage of our algorithms over short time windows, rather than across the entire time horizon. Specifically, we report the local coverage using a window size of 100, following the approach adopted in [2]. Based on the results, both of our proposed algorithms, GMOCP and EGMOCP, achieve local coverage close to the target level. This result validates the ability of our methods to quickly adapt to abrupt distribution shifts.
>
> **RC6 Elaborate online conformal prediction through the introduction:** Thank you for the suggestion. We have revised the introduction to provide a more complete overview of the online conformal prediction (OCP) framework.
>
> [1] Gibbs, I. and Candes, E., 2021. Adaptive conformal inference under distribution shift. Advances in Neural Information Processing Systems, 34, pp.1660-1672.
>
> [2] Bhatnagar, A., Wang, H., Xiong, C. and Bai, Y., 2023, July. Improved online conformal prediction via strongly adaptive online learning. In International Conference on Machine Learning (pp. 2337-2363). PMLR.
>
> [3] Angelopoulos, A.N., Barber, R. and Bates, S., 2024, July. Online conformal prediction with decaying step sizes. In International Conference on Machine Learning (pp. 1616-1630). PMLR.
>
> [4] Gibbs, I. and Candès, E.J., 2024. Conformal inference for online prediction with arbitrary distribution shifts. Journal of Machine Learning Research, 25(162), pp.1-36.

---

### Review · Reviewer_dmgy · 2025-07-22

**Summary Of Contributions:**

The paper focuses on the problem of adaptive online conformal prediction, where at each time step conformal prediction is deployed using a model chosen from a set of candidate models. The authors propose a dynamic graph-based selection schema to select the candidate set of models as well as the final model and its adaptive coverage probability at each time step. This schema consists of a bipartite time evolving graph representing multiple selections of models that can be chosen as the candidate set. The authors construct each selection of models as well as choose the final model using probabilistic rules that are updated per time step based on the realized coverage error. In addition, they suggest an extension of their update rule to encourage smaller prediction sets. The authors provide theoretical bounds on the coverage error and regret of both variants of their method and evaluate them against competitive baselines on several datasets.

**Audience:**

Yes

**Broader Impact Concerns:**

No concerns.

**Claims And Evidence:**

Yes

**Requested Changes:**

**Definitions/Theory**
1. In theorem 1,  the values of $\eta$ seem to be positive according to the proof in appendix A.1, where $\eta$ is the learning rate as defined in eq. 5. It would be helpful to clarify this on Theorem 1 as well (e.g., by adding a clarification that $\eta > 0$) and also clarify what would mean--both intuitively and in practice--to select an initial miscoverage rate  that is negative, i.e., $\alpha_{1}^{m} = -\eta$ for some model $m$. Similarly, it would be helpful to clarify what would it mean to allow for a miscoverage rate that is larger than $1$ by setting $\alpha_{1}^{m} = 1 + \eta$.  The same applies for theorem 3. (critical)
2. The authors use $K$ for the number of labels as well as in the proof of theorem 1. Using a different parameter would prevent ambiguity. (minor)
3. The term $b_{m,q}$ first introduced above theorem 1, seems to be first defined below eq. 46 in the Appendix. This is rather confusing and the authors should either point to the appendix whenever $b_{m,q}$ is first introduced or define it inline. (minor)


**Typos/Clarifications**

1. It would be important to clarify the train/test splits used for the CIFAR  datasets. (critical)
2. In page 3 end of introduction in section 3 there are double ‘.’ (minor)
3. Above equation 6, the authors perhaps could write ‘The parameter $\eta$’ instead of directly writing ‘. $\eta$’ (minor)
4. The authors should define epsilon in equation 7 (minor)
5. In section 3.1 the authors should explain the acronyms they are using (e.g. PMF, SF-OGD) the first time they introduce them (minor)
6. Throughout the manuscript ‘Miss coverage’ —> ‘miscoverage’ (minor)
7. Theorem 3 typo ‘$C-1$’ —> $C_1$ (minor)
8. In the caption of Table 1 there is a duplicate comment on the significance of bold text. (minor)

**Suggestions for improvement**

1. The title looks a bit hard to read. Perhaps ‘multimodel online conformal prediction with graph-structured feedback’ would be a bit easier to parse.
2. In the experimental evaluation, the authors may like to present results on the same dataset
3. In Figure 2, it would be helpful to show also the average size of the other baselines to be able to see the advantage of the proposed method.

**Strengths And Weaknesses:**

**Strengths**

The dynamic selection of the candidate set of models based on the online feedback seems quite interesting as it seems more flexible than prior methods, and able to adapt efficiently the candidate set based on the observed coverage error.  The work is well organized, clearly motivated and  appropriately placed into contemporary literature. The paper seems nicely written and clear. The theoretical and experimental results seem to sufficiently  support the proposed method.

**Weaknesses**

In theorems 1 and 3 seems to allow for a negative initial miscoverage rate that seems somewhat counterintuitive, while it is unclear how is this applicable in practice. There are several typos and a few undefined terms throughout the manuscript (please refer to changes for details).

---

> ### Author Response · Authors · 2025-08-09
>
> We thank the reviewer for finding our work interesting, clearly motivated, and well-situated within the contemporary literature. Please find our responses to the Requested Changes(RC) below.
>
> **RC1 Definition/Theory:**
>
> **RC1.1 Add $\eta>0$ and clarify the term $\alpha_1^m = -\eta$ in Theorems 1 and 3:**     We thank the reviewer for the careful and thoughtful feedback. We have now clarified in Theorems 1 and 3 that $\eta > 0$, and we appreciate the suggestion to make this explicit.
> Regarding the term $\alpha_1^m \in [-\eta, 1+ \eta]$, we note that its inclusion was unnecessary and has been removed from the revised theorems. This initialization is not required to establish the coverage guarantee. In our implementation, we initialize $\alpha_1^m = \alpha, \hspace{0.1cm} \forall m \in [M]$. The interval $\alpha_t^m \in [-\eta, 1+ \eta]$ is required for the regret analysis (see Equation 35). This bound is proved in Lemma 2 by [1]. Intuitively, due to the adaptive nature of the miscoverage update rule (Equation 5), $\alpha_t^m$ may temporarily take values slightly outside the $[0,1]$ interval. However, in practice, we apply a projection step to clip $\alpha_t^m$ into the valid range $[0,1]$, ensuring that prediction sets remain meaningful and consistent with conformal prediction guarantees. We have removed the unnecessary sentence about the initial miscoverage setting from both Theorems 1 and 3 in the revised version. We appreciate the reviewer’s detailed comments, which helped us clarify these theoretical claims.
>
> **RC1.2 Select a different parameter for number of labels to prevent ambiguity:** To address this comment, the notation $K$ has been replaced with $N_{labels}$ to denote the total number of labels. In the revised version, the notation $K$ is only used in the proof of Theorem 1.
>
> **RC1.3 Introduce notation $b_{m,q}$:** Thank you for your suggestion. We have added a sentence ``see Appendix A.2 for full definition of $b_{m,q}$'' where the notation $b_{m,q}$ first appears in the main text, directing readers to the appendix for its full definition.
>
> **RC2 Typos/Clarification:** We thank the reviewer for pointing out the typos and suggesting clarifications. We have clarified the data split in the experimental section of the revised paper: each dataset is divided into a training phase (50,000 samples) and a test set (6,000 samples). Additionally, a separate set of 2,000 samples is used for parameter selection in the conformal prediction task. We have also addressed all remaining minor issues noted by the reviewer in the revised manuscript.
>
>   **RC3 Suggestions for improvement:**
>      We would like to thank the reviewer for their detailed comments to help improve our manuscript. Below, we detail the corresponding updates in the revised version.
>
> **RC3.1 Update the title:** Thank you for your suggestion. We find your suggested title indeed would be a better fit for our paper and would read better. Hence, we have updated the title accordingly. We appreciate your help in improving the readability of our title.
>
> **RC3.2 Include results on the same dataset:** Thank you for the suggestion. To address this comment, we included additional numerical results on the same dataset (CIFAR-10C) in Section B.9 for both sudden and gradual distribution shifts. It can be observed that in both settings, our proposed method, GMOCP, leads to lower runtime, and the EGMOCP approach results in prediction sets with lower average width and higher single width.
>
> **RC3.3 Average set size of baselines over time:** We have included an additional experiment in the revised version to compare the average prediction set sizes generated by each algorithm over the full time horizon (see Appendix B.8). EGMOCP achieves the smallest average prediction set size among all benchmarks, while COMA exhibits significant fluctuations, resulting in larger prediction sets at certain timesteps.
>
>
> [1] Hajihashemi, E. and Shen, Y., 2024. Multi-model ensemble conformal prediction in dynamic environments. Advances in Neural Information Processing Systems, 37, pp.118678-118700.

---

### Review · Reviewer_QDNW · 2025-07-26

**Summary Of Contributions:**

This paper studies online conformal prediction, where a prediction set is generated to ensure the desired coverage rate of the true label. It focuses on the case where there is a set of candidate models with varying qualities. The main contribution of the paper is a method that efficiently selects a model for constructing the prediction set, based on a bipartite graph and carefully designed feedback. Regret analysis shows that the proposed method can achieve the desired coverage guarantee, and experimental results validate its effectiveness.

**Audience:**

Yes

**Claims And Evidence:**

No

**Requested Changes:**

* Could you provide more evidence on how the proposed method is computationally efficient and can scale to situations where $M$ is large? (See the first point in the weaknesses section for more details.)
* Could you discuss how $N$ and $J$ affect the theoretical guarantees of the proposed method?
* Why does EGMOCP have lower running time for $J = 4$ than for $J = 2$?
* How did you choose the parameters used in the experiments?

**Strengths And Weaknesses:**

**Strengthes:**
- Conformal prediction with multiple models is an interesting and relevant problem, especially as the use of diverse model ensembles becomes increasingly common. Proposing a method that improves computational efficiency is valuable, as it enables scaling to incorporate a larger number of models.
- The paper provides a theoretical analysis of the coverage rate, supporting the validity of the proposed method.
- The manuscript is well-structured.

**Weaknesses:**
- On computational efficiency: One of my main concerns about the paper is the significance of the computational improvement achieved by the proposed methods. As stated in Section 3.2, the computational cost of the GMOCP method is $O(MNJ)$. It is unclear to me how substantial the improvement is, given that the cost scales linearly with the number of candidate models $M$, and may still become prohibitive when $M$ is large.
- About the Theorem 1: It seems to me that some parameters in the theorems are not clearly specified. For example, it is somewhat surprising that the bound appears to be independent of $J$ and $N$. Does this imply that the choices of $J$ and $N$ do not affect the coverage and regret guarantees of the proposed method? In addition, some algorithmic parameters are not well specified. For instance, is a particular design or selection of $\epsilon$ (and $\beta$ in Theorem 3) required to achieve the stated bounds?
- About the experiments: The experiments are a bit counterintuitive to me. Regarding the running time of the EGMOCP method, the case with $J = 4$ shows even lower running time than $J = 2$, which does not align with the $O(MNJ)$ computational cost stated in the analysis. Could you explain why this is the case?
- In Section 4.1, the paper mentions that "the parameters $epsilon$, $eta$, and $beta$ are selected through grid search...". I am wondering what data was used to perform the grid search. Since the problem is set in an online learning context, it is unclear whether an additional validation set was available for this purpose.

---

> ### Author Response · Authors · 2025-08-09
>
> We sincerely appreciate the reviewer's constructive and insightful comments, which have helped us improve the quality of our work. We are pleased that the reviewer found our proposed approach ``valuable for improving computational efficiency ''. Please find our responses to the weaknesses (W) and requested changes (RC) below.
>
> **W1+RC1 Computational efficiency and how the approach scales when the total number of models, $M$, is large:** We thank the reviewer for raising the question about the computational complexity of the proposed methods. The per-iteration cost at time $t$ for GMOCP is $O(Nt + JMN)$, where the $JMN$ term accounts for graph generation and the $Nt$ term arises from computing $\bar{\alpha}_t^m$ for up to $N$ selected models, using sorted calibration scores of length $t$. The per-iteration complexity of MOCP is $O(Mt)$. Thus, GMOCP can reduce the per-iteration complexity compared to MOCP, especially in settings where $N << M$. Furthermore, note that while addressing your question, we found that there was a typo regarding the per-iteration computational complexity of GMOCP, which we have corrected in the updated version of the paper. The computational complexity of the GMOCP method should be $O(Nt + JMN)$ instead of $O(JMN)$.
>
>  **W2+RC2 The impact of $N$ and $J$ on theoretical guarantees and algorithmic parameter specifications:** The parameters $N$ and $J$ do affect the theoretical guarantees. Specifically, the coverage bound in Theorem 1 depends on constants $B_1$ and $B_2$, where $B_1$ is a function of $N$ and $b = \lfloor log_2^J\rfloor$, and $B_2$ is a function of $N$; see Lemma 3 for their full definitions. Moreover, the term $b$ also appears in the regret bound in Theorem 2, which reflects the influence of $J$ on the regret bound. Regarding algorithmic parameters, they were clarified in Appendix A.1 in the submitted manuscript. Specifically, to establish the coverage guarantee for GMOCP (Theorem 1), parameter $\epsilon$ should be set as $\epsilon = T^{-\frac{3}{4}}$, and the same setting is used for Theorem 3. For the regret bound in Theorem 2, it can be seen from Appendix A.2 in the original submission that the parameter $\epsilon$ should be set as $\epsilon=\frac{1}{\sqrt{T}}$. Finally, to achieve the regret bound for EGMOCP in Theorem 4, we specified at the end of Appendix A.4 in the submitted manuscript that we set $\beta = \frac{1}{\sqrt{T}}$.
>
> **W3+RC3 Counterintuitive run time metric:**
>     We thank the reviewer for this sharp observation, which helps us to realize the potential inconsistency in experimental environments (specifically, GPU workload) when running experiments for different parameters. Since the GPU was concurrently used by multiple users while running experiments, this may have led to inconsistencies in runtime measurements. To validate and address this issue, we re-run the experiments under controlled conditions, ensuring that the GPU was exclusively allocated to our experiment. The results have been updated accordingly in the revised version.  We observed that the runtime results change, while all other experimental outcomes remain unchanged from the previous experiments. It can be observed from the revised tables (e.g., Table 1) that GMOCP still demonstrates improved runtime performance compared to all benchmarks, same as before. The counterintuitive situations where EGMOCP shows `` lower runtime for \(J=4\) than for \(J=2\)'' no longer appear in the new results. Based on this observation, we believe the counterintuitive runtime results are likely due to the different GPU workload when running the previous experiments. We very much appreciate the reviewers' sharp observation, which has helped us resolve the inconsistent results.

---

> ### Author Response · Authors · 2025-08-09
>
> **W4+RC4 Parameter selection procedure:** We thank the reviewer for the question regarding the parameter selection procedure. To select these parameters, we used an additional data split of 2000 samples for parameter selection. We considered the candidate sets \{1/10,1/20,1/30\} and \{0.4,0.5,0.6\} for $\eta$ and $\epsilon$, respectively, for the GMOCP algorithm $(N=1,J=1)$ on the CIFAR-100C dataset under a sudden distribution shift. This resulted in 9 different configurations. While the goal is to construct smaller prediction sets, we also ensure that coverage is not significantly compromised. To this end, for each of the 9 configurations, we compute both the coverage and the average prediction set size. We then set a threshold equal to the average coverage across all configurations and select, among the settings that exceed this threshold, the one with the smallest prediction set size. After determining the appropriate values for $\eta$ and $\epsilon$, we fixed these parameters for all experimental settings.  We would like to emphasize that such a parameter selection procedure is suboptimal, as the results could likely be improved if the grid search for parameter selection were performed separately for each experimental setting instead of only for GMOCP $(J=1, N=1)$ with CIFAR-100C under sudden distribution shift. Hence, the performance of the proposed methods has room to improve if a more specific parameter selection procedure is employed. For selecting the value of $\beta$, we considered the candidate set \{0.01,0.02,0.03,0.04,0.05\}, and conducted experiments on the CIFAR-100C dataset under a sudden distribution shift using the EGMOCP algorithm $(J=1, N=1)$. The procedure for selecting the $\beta$ followed the same criteria used for $\eta$ and $\epsilon$; we computed coverage and average prediction set size for each setting and set the coverage threshold to the average coverage across all settings. Among the configurations exceeding this threshold, we selected the one with the smallest prediction set size.

---

### Author Response · Authors · 2025-09-11
**Follow up**

Dear Reviewers and Action Editor,

We would like to sincerely thank you again for your constructive comments and suggestions on our manuscript. We hope that our responses have addressed your concerns.

We are writing to kindly follow up regarding the status of the manuscript. If there are any remaining questions or points that require further clarification, we would be more than happy to address them.

Thank you very much for your time and consideration.

Best regards,

Authors of submission 5156

---

### Decision · Action_Editor_yqwY · 2025-10-06

**Recommendation:** Accept as is

**Audience:**

Yes

**Audience Explanation:**

I would expect that this paper would be of interest to some researchers working on conformal prediction.

**Claims And Evidence:**

Yes

**Claims Explanation:**

The paper focuses on the problem of adaptive online conformal prediction, where at each time step conformal prediction is deployed using a model chosen from a set of candidate models. The authors propose a dynamic graph-based selection schema to select the candidate set of models as well as the final model and its adaptive coverage probability at each time step. Although the reviewers originally had concerns about the claims being supported by accurate, convincing and clear evidence, they were satisfied with the authors' response.
- Reviewer djnH was originally concerned that some of the claims in this paper were not well-supported (e.g., lack of theoretical analysis to support computational efficiency claims and lack of empirical comparison to some existing baselines). The authors authors added more experiments and theoretical results, which alleviated the reviewer's concerns.
- Reviewer dmgy found some of the theoretical claims to be unclear, but was satisfied with the authors' clarifications and revisions.
- Finally, Reviewer QDNW had concerns about inconsistencies between the theoretical analysis and empirical results, but was satisfied by the authors' feedback. I'll note that they still have reservations about the parameter selection, writing that "since the work focuses on an online learning setting, requiring an additional validation set appears to be a rather strong assumption."